# Anillin regulates epithelial cell mechanics by structuring the medial-apical actomyosin network

Torey R Arnold[1], Joseph H Shawky[2,3,4], Rachel E Stephenson[1], Kayla M Dinshaw[1], Tomohito Higashi[1†], Farah Huq[1], Lance A Davidson[3,4], Ann L Miller[1]*

[1]Department of Molecular Cellular and Developmental Biology, University of Michigan, Ann Arbor, United States; [2]Department of Bioengineering, University of Pittsburgh, Pittsburgh, United States; [3]Department of Developmental Biology, University of Pittsburgh, Pittsburgh, United States; [4]Department of Computational and Systems Biology, University of Pittsburgh, Pittsburgh, United States

**Abstract** Cellular forces sculpt organisms during development, while misregulation of cellular mechanics can promote disease. Here, we investigate how the actomyosin scaffold protein anillin contributes to epithelial mechanics in *Xenopus laevis* embryos. Increased mechanosensitive recruitment of vinculin to cell–cell junctions when anillin is overexpressed suggested that anillin promotes junctional tension. However, junctional laser ablation unexpectedly showed that junctions recoil faster when anillin is depleted and slower when anillin is overexpressed. Unifying these findings, we demonstrate that anillin regulates medial-apical actomyosin. Medial-apical laser ablation supports the conclusion that that tensile forces are stored across the apical surface of epithelial cells, and anillin promotes the tensile forces stored in this network. Finally, we show that anillin's effects on cellular mechanics impact tissue-wide mechanics. These results reveal anillin as a key regulator of epithelial mechanics and lay the groundwork for future studies on how anillin may contribute to mechanical events in development and disease.
DOI: https://doi.org/10.7554/eLife.39065.001

*For correspondence:
annlm@umich.edu

Present address: †Department of Basic Pathology, Fukushima Medical University, Fukushima, Japan

Competing interests: The authors declare that no competing interests exist.

## Introduction

During development, an organism takes its shape by generating forces and establishing mechanical properties at the cellular level (*Davidson, 2012*). Forces produced by the contractile actomyosin cytoskeleton within cells are transmitted between cells and through the tissue via cell–cell junctions. By coordinating which cells are contracting, elongating, or rearranging, tissues can bend, fold, or elongate allowing for the complex tissue organization found in many multicellular organisms. Understanding how cell-scale mechanical inputs result in tissue-scale changes requires a comprehensive knowledge of the proteins involved in controlling cellular mechanics.

Anillin is a scaffolding protein that was first characterized for its role in cell division (*Field and Alberts, 1995*) and has since been shown to regulate cytokinesis in organisms ranging from yeast to humans (*Piekny and Maddox, 2010*). Anillin localizes to the actomyosin contractile ring during cytokinesis and primarily to the nucleus during interphase (*Field and Alberts, 1995*; *Oegema et al., 2000*; *D'Avino, 2009*). During cytokinesis, anillin anchors the contractile ring to the plasma membrane by binding to actomyosin through its N-terminal myosin- and F-actin-binding domains and to lipids through its C-terminal C2 and PH domains (*Straight et al., 2005*; *Piekny and Glotzer, 2008*; *Liu et al., 2012*; *Sun et al., 2015*). Anillin can both enhance and limit contractility during cytokinesis through binding directly to active RhoA as well as several proteins that positively or negatively

regulate Rho, and by crosslinking F-actin (*Piekny and Glotzer, 2008*; *Frenette et al., 2012*; *Manukyan et al., 2015*; *Descovich et al., 2018*).

Recently, novel roles for anillin in regulating epithelial cell–cell junctions have been reported. Our group showed that anillin is localized to cell–cell junctions in frog embryos where it regulates cell–cell junction integrity, as both tight junctions and adherens junctions were disrupted when anillin was knocked down (*Reyes et al., 2014*). Furthermore, we showed that anillin scaffolds the contractile actomyosin machinery connected to epithelial cell–cell junctions (*Reyes et al., 2014*). Proteomic studies supported anillin's association with cadherin-mediated adhesions in fly and human cells (*Guo et al., 2014b*; *Toret et al., 2014*). Studies in human epithelial cells confirmed and expanded our understanding of anillin's function at cell–cell junctions. Anillin knockdown in cultured human epithelial cells resulted in junction disassembly and disorganized junctional F-actin and myosin II (*Wang et al., 2015*). A recent study reported that myosin II anchors anillin to the apical cell cortex, and cycles of binding and unbinding to anillin stabilize active RhoA by increasing its cortical residence time, thus promoting junctional actomyosin contractility (*Budnar et al., 2018*). Finally, in vivo studies in zebrafish demonstrated that barrier function in the kidney, which is mediated by tight junctions, was compromised by a mutation in anillin that causes Focal Segmental Glomerulosclerosis, a leading cause of kidney failure (*Gbadegesin et al., 2014*).

Despite these emerging roles for anillin in regulating actomyosin-mediated cell–cell junction contractility, very little is known about how anillin contributes to the mechanical properties of epithelial tissues in development and disease. Notably, anillin is expressed during developmental events that are controlled by mechanical inputs from cells and tissues, including gastrulation and neurulation (*Session et al., 2016*). Anillin is also overexpressed in multiple human cancers (*Hall et al., 2005*). Increased tissue stiffness is frequently observed during tumorigenesis; tumor growth and metastases are sensitive to mechanical cues, which can significantly affect cancer prognosis (*Kumar and Weaver, 2009*). Changes in mechanical properties can be mediated by changes in the composition of the extracellular matrix as well as changes in the organization of the intracellular cytoskeleton. Although there is little research on how anillin may contribute to the mechanical properties of cancer cells, it is becoming clear that anillin's function extends beyond cytokinesis. Anillin resides not only in the nucleus of interphase cancer cells, but also in the cytoplasm, and anillin expression can have either positive or negative outcomes for patients depending on the cancer type and anillin's subcellular localization (*Hall et al., 2005*; *Suzuki et al., 2005*; *Ronkainen et al., 2011*; *Liang et al., 2015*; *Magnusson et al., 2016*; *Wang et al., 2016*; *Idichi et al., 2017*; *Zhang et al., 2018*).

Based on anillin's demonstrated role in regulating cell–cell junction contractility and potential involvement in regulating development and disease, we sought to investigate whether anillin affects epithelial cell and tissue mechanics. We used the animal cap epithelium of developing *Xenopus laevis* embryos as a model vertebrate epithelial tissue. Using a combination of techniques including live imaging, laser ablation, and tissue stiffness measurements, we identified a new role for anillin in organizing F-actin and myosin II at the medial-apical surface of epithelial cells. We show that anillin promotes a contractile medial-apical actomyosin network, which produces tensile forces in individual cells that are transmitted between cells via cell–cell junctions to promote tissue stiffness.

## Results

### Anillin increases junctional vinculin recruitment but reduces recoil of junction vertices after laser ablation

Since anillin can both promote and limit contractility at the cytokinetic contractile ring (*Piekny and Glotzer, 2008*; *Manukyan et al., 2015*; *Descovich et al., 2018*), and anillin localizes to cell–cell junctions where it maintains F-actin, myosin II, and proper active RhoA distribution (*Reyes et al., 2014*), we sought to test whether anillin affects junctional tension. As a readout of relative tension on junctions, we quantified the junctional accumulation of Vinculin-mNeon. High junctional tension induces a conformational change in α-catenin, which recruits vinculin to adherens junctions to reinforce the connection to the actin cytoskeleton (*Yonemura et al., 2010*). We have previously vetted a tagged vinculin probe in *Xenopus laevis* and used it to show that the cytokinetic contractile ring applies increased tension locally on adherens junctions (*Higashi et al., 2016*). To test how anillin affects junctional vinculin recruitment, we knocked down anillin with a previously characterized morpholino

oligonucleotide (*Reyes et al., 2014*) or overexpressed a tagged version of anillin (Anillin-3xmCherry). In control embryos, Vinculin-mNeon accumulated weakly along bicellular junctions and strongly at tricellular contacts, which are sites of higher tension (*Figure 1A*) (*Choi et al., 2016*; *Higashi and Miller, 2017*). When anillin was knocked down, the intensity of Vinculin-mNeon was reduced at junctions (*Figure 1A,B*). We also examined mCherry-α-catenin intensity because vinculin intensity could vary based on the amount of its binding partner α-catenin available at junctions (*Figure 1—figure supplement 1A,C*). When anillin was knocked down, mCherry-α-catenin was strongly reduced at junctions; therefore, the reduced Vinculin-mNeon intensity may not indicate reduced tension because α-catenin is also reduced (*Figure 1—figure supplement 1A–C*). Notably, when anillin was overexpressed, the intensity of Vinculin-mNeon at junctions was significantly elevated (*Figure 1A,B* and *Figure 1—figure supplement 1A,B*), and the intensity of mCherry-α-catenin was unchanged (*Figure 1—figure supplement 1A,C*), indicating that the increased Vinculin-mNeon recruitment was due to a tension-induced conformation change in α-catenin.

Changes in cell shape also supported the conclusion that anillin affects junctional tension. When anillin was knocked down, cells became less polygonal and more rounded (*Figure 1A*), a hallmark of reduced tension (*Yonemura et al., 2010*). This is consistent with our previous report that anillin knockdown leads to effects associated with reduced tension including cell rounding, apical doming, and loss of F-actin and myosin II from junctions (*Reyes et al., 2014*). In contrast, when anillin was overexpressed, junctions exhibited a wavy, distorted shape compared to more linear control junctions (*Figure 1A*). Interestingly, wavy junctions can indicate either unbalanced high tension (*Tokuda et al., 2014*; *Nowotarski and Peifer, 2014*) or uniform low tension along the junction (*Nowotarski and Peifer, 2014*; *Choi et al., 2016*).

As a complementary approach to test whether anillin promotes junctional tension, we measured junction recoil after laser ablation. The amount of junction recoil (measured as the distance between cell vertices over time) is reported to correlate with the relative junctional tension such that a junction under high tension will exhibit enhanced recoil (*Farhadifar et al., 2007*; *Fernandez-Gonzalez et al., 2009*). We used a 2-photon laser to ablate a local site in the center of a bicellular junction in embryos expressing E-cadherin-3xGFP. To avoid potential contributions of junction length to recoil, junctions of a similar length were selected for ablation in control, anillin knockdown, and anillin overexpressing embryos (*Figure 1E*). After laser ablation, control junctions recoiled 2.8 ± 0.2 µm after 18 s (*Figure 1C,D*, *Figure 1—figure supplement 1D*, and *Video 1*). Based on our hypothesis that anillin promotes junctional tension, we expected to see reduced junction recoil when anillin was knocked down; however, recoil increased to 4.1 ± 0.4 µm after 18 s (*Figure 1C,D*, *Figure 1—figure supplement 1D*, and *Video 1*). When anillin was overexpressed, we expected to see increased recoil; however, the recoil decreased to 1.6 ± 0.4 µm after 18 s (*Figure 1C,D*, *Figure 1—figure supplement 1D*, and *Video 1*). A similar trend in recoil dynamics was observed at later time points. After 45 s (the last time point where we could accurately track the junction vertices), control cells and anillin knock down cells reached a similar amount of recoil, while anillin overexpressing cells recoiled significantly less than controls (*Figure 1—figure supplement 1E*).

Taken together, the vinculin recruitment data indicated that anillin increased junctional tension, whereas the laser ablation data showed that anillin reduced junction recoil, suggesting reduced junctional tension. Although the vinculin recruitment and laser ablation results initially seemed to be at odds with one another, an interesting phenotype emerged from the laser ablation data when we observed the Anillin-3xmCherry channel in anillin overexpression embryos, as anillin highlights the apical surface of the cells as well as the cell–cell junctions. In multiple instances, we saw that the two cells adjacent to the junction that was ablated dramatically lost adhesion and recoiled *perpendicular* to the junction instead of in the expected direction *parallel* to the junction (*Figure 1F*, *Figure 1—figure supplement 2A*, and *Video 2*). Using E-cadherin 'splaying' (meaning that the stubs of the junction left after laser ablation were separated in two) as a readout of *perpendicular* separation, we found that anillin overexpressing cells separate perpendicularly slightly, but not significantly, more often than control cells, whereas perpendicular separation of anillin knockdown cells is dramatically reduced compared to controls (*Figure 1—figure supplement 2B,C*). This perpendicular separation results led us to hypothesize that anillin was somehow responsible for exerting forces *perpendicular* to the junction, possibly through the medial-apical actomyosin cortex. In this way, anillin could both increase vinculin intensity at junctions and reduce recoil in the expected *parallel* direction after laser ablation.

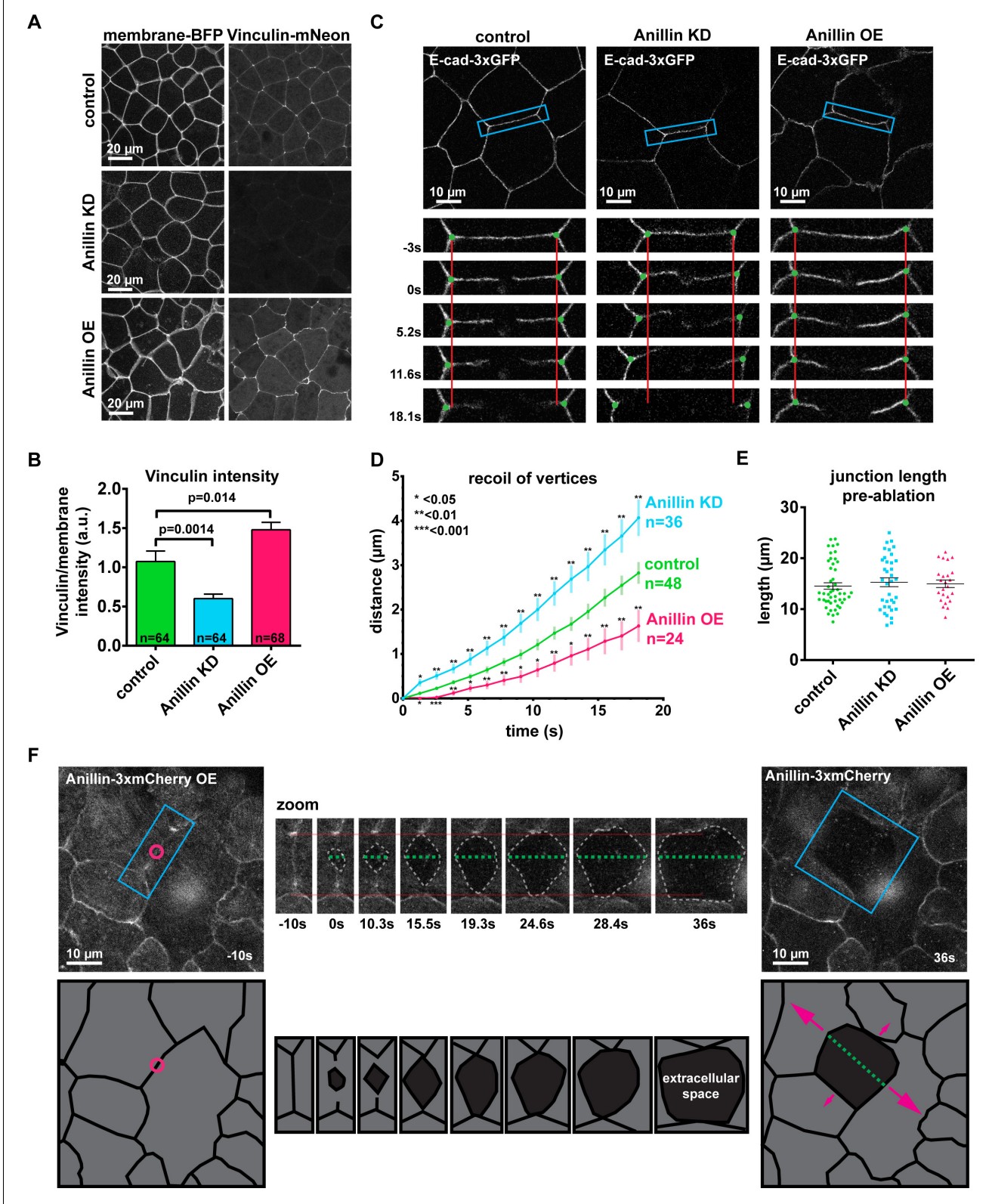

**Figure 1.** Anillin increases junctional vinculin recruitment but reduces recoil of junction vertices after laser ablation. (**A**) Confocal images of live epithelial cells in gastrula-stage *Xenopus laevis* embryos expressing a probe for the plasma membrane (2x membrane localization signal of Src family tyrosine kinase Lyn tagged with TagBFP at its C-terminus) and inculin-mNeon when anillin is knocked down (KD) or overexpressed (OE). (**B**) Quantification of vinculin intensity as a ratio to membrane intensity. Measurements were taken by tracing a bicellular junction from vertex to vertex. Error bars, S.E.M.
*Figure 1 continued on next page*

*Figure 1 continued*

Statistics, unpaired t-test, n = number of junctions. (C) Cell view images showing E-cadherin tagged with 3xGFP prior to junctional laser ablation. Blue boxes show the zoomed area for the ablation montage. Vertical red lines indicate the initial location of junction vertices; green dots indicate the location of junction vertices measured after ablation. (D) Quantification of junction vertex separation over time after laser ablation (at time 0 s). Error bars, S.E.M. Statistics, unpaired t-test, n = number of junctions. (E) Quantification of initial junction length from vertex to vertex prior to ablation. Error bars, S.E.M. Unpaired t-tests between control and anillin KD or control and anillin OE resulted in no statistical significance; Control vs. KD, p=0.49, Control vs OE, p=0.69. (F) Cell view of embryo overexpressing anillin tagged with 3xmCherry. Blue boxes show the zoomed area for the ablation montage. Horizontal red lines indicate the initial location of junction vertices; green dashed line indicates the perpendicular separation between the two cells, and the gray dashed outline indicates the extracellular space forming between the two cells. Cartoon traces of the ablation data depicting the events during ablation are shown below. Pink circle indicates the site of laser ablation, green dashed line indicates perpendicular separation between cells, and pink arrows represent direction and magnitude of forces where larger arrows represent larger forces. (Unannotated data can be found *Figure 1—figure supplement 2A*).

DOI: https://doi.org/10.7554/eLife.39065.002

The following source data and figure supplements are available for figure 1:

**Source data 1.** Source data for *Figure 1B,D,E*, *Figure 1—figure supplement 1B,C,E* and *Figure 1—figure supplement 2C*.

DOI: https://doi.org/10.7554/eLife.39065.005

**Figure supplement 1.** Anillin increases junctional vinculin recruitment but reduces recoil of junction vertices after laser ablation.

DOI: https://doi.org/10.7554/eLife.39065.003

**Figure supplement 2.** Anillin increases perpendicular junction separation after junctional laser ablation.

DOI: https://doi.org/10.7554/eLife.39065.004

## Anillin structures medial-apical F-actin and myosin II

Based on our finding that anillin promotes forces acting perpendicular to the junction, we next tested whether anillin affects the medial-apical populations of F-actin and myosin II, which could generate these perpendicular forces. To observe medial-apical F-actin, we optimized our fixation conditions to preserve this delicate network of F-actin and stained it with phalloidin. In control cells, we observed a dense meshwork of thin filaments of medial-apical actin. Knocking down anillin reduced the intensity of this network, while overexpressing anillin drastically reorganized this network into thick bundles of F-actin that could span the entire surface of the cells (*Figure 2A,E*). In live cells overexpressing anillin, we also observed thick medial-apical bundles of F-actin using a probe for F-actin (Lifeact-GFP), and Anillin-3xmCherry colocalized with the F-actin bundles (*Figure 2B* and *Video 3*). Varied F-actin bundle organization and dynamics were observed including linear arrays that spanned the apical surface of the cells, aster-like structures, and circular bundles that rotated or flowed from the junction towards the center of the cell (*Figure 2A,B* and *Video 3*).

To determine whether these medial-apical F-actin bundles were decorated with myosin II, we performed live imaging with a probe for myosin II, the SF9 intrabody (*Nizak et al., 2003*; *Vielemeyer et al., 2010*; *Hashimoto et al., 2015*), which we tagged with mNeon. In controls, myosin II accumulated on the medial-apical surface in patches that resemble previously described cortical waves of F-actin (*Bement et al., 2015*) (*Figure 2C*). Knocking down anillin resulted in almost a complete loss of myosin II from the apical surface of epithelial cells, and when anillin was overexpressed, myosin II was increased and decorated the large bundles of F-actin that formed (*Figure 2C,D,F*). These data identify a new role for anillin in establishing and structuring proper medial-apical F-actin and myosin II.

## Anillin increases tensile forces and contractile energy stored across the medial-apical surface of epithelial cells

Based on our findings that anillin promotes and structures actomyosin accumulation across the

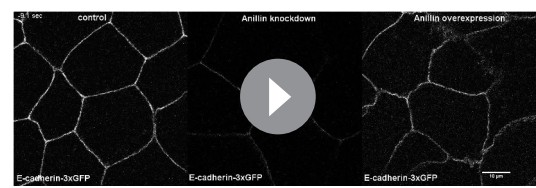

**Video 1.** Increased anillin expression reduces vertex separation following junctional laser ablation. Cell views of E-cadherin-3xGFP in control, anillin knockdown, or anillin overexpressing *Xenopus laevis* embryos before and after junctional laser ablation. Ablation was performed at time 0 s. For the anillin knockdown video, note that the reduction of E-cadherin intensity over time is due to drift of the sample out of the focal plane.

DOI: https://doi.org/10.7554/eLife.39065.006

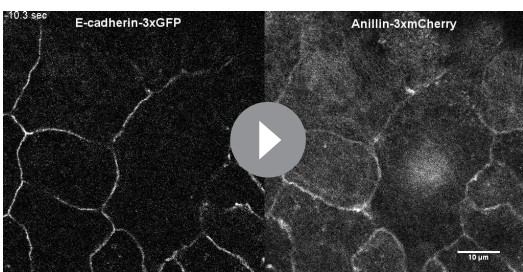

**Video 2.** Example of recoil perpendicular to the junction following junctional laser ablation when anillin is overexpressed. Cell views of a *Xenopus laevis* embryo with anillin overexpressed before and after junctional laser ablation. Probes are E-cadherin-3xGFP and Anillin-3xmCherry. Note that there is minimal recoil in the direction parallel to the ablated junction, but significant recoil in the perpendicular direction.
DOI: https://doi.org/10.7554/eLife.39065.007

apical surface of cells, we examined whether tensile force was generated and contractile energy was stored in this network. Building on our observation that when anillin was overexpressed, some cells dramatically separated perpendicularly to the junction after laser ablation (*Figure 1F*), we tested whether the large bundles of actomyosin that form upon anillin overexpression exert contractile force on junctions. Using junctional vinculin intensity as a readout of contractile forces exerted on junctions (*Higashi et al., 2016*; *Hara et al., 2016*), we found that junctions associated with medial-apical F-actin bundles connected *perpendicularly* to the junction exhibited increased recruitment of vinculin compared to junctions associated with medial-apical bundles that run *parallel* to the junction (*Figure 3A,B*).

We then probed the tensile forces stored across the medial-apical surface by making laser cuts of medial-apical F-actin (*Saha et al., 2016*; *Ma et al., 2009*) (*Figure 3C–G* and *Video 4*). We measured junction recoil (apical expansion) in two directions, *perpendicular* to the cut site and *parallel* to the cut site (*Figure 3C–E*). After laser ablation of medial-apical actin (Lifeact-RFP), control cells opened 3.0 ± 0.2 µm *perpendicular* to the cut site after 18 s. Anillin overexpressing cells recoiled significantly more (5.1 ± 0.4 µm), indicating that the medial-apical actin in these cells stores more tensile force, whereas anillin knockdown cells recoiled less than controls (2.0 ± 0.2 µm) in the perpendicular direction (*Figure 3C,D*, and *Video 4*). When apical expansion *parallel* to the cut site was measured, control, anillin knockdown, and anillin overexpressing cells all opened similar distances after laser ablation of medial-apical F-actin (*Figure 3C,E*, and *Video 4*). Initial apical cell sizes prior to ablation were similar; anillin knockdown cells were slightly but significantly larger than controls, while anillin overexpressing cells were not significantly different from controls (*Figure 3—figure supplement 1A, B*), suggesting that the differences in *perpendicular* expansion after medial-apical laser ablation, weren't simply a consequence of apical cell size. Additionally, control, anillin knockdown, and anillin overexpressing cells exhibited similar cell geometry when the aspect ratios of the cells were compared (*Figure 3—figure supplement 1C*).

In addition to tracking apical cell expansion in two directions, we aimed to more robustly track the changes occurring locally in the ablated cell and the surrounding tissue by measuring the strain released after medial-apical F-actin ablation. To do this we used software that tracked individual pixels in images from the first and last time point after laser ablation. The shapes of cells within these images were transformed to fit one another, and the distance each pixel had to be moved to fit the two images can be mapped into a strain area intensity for each pixel (*Feroze et al., 2015*). The area strain map indicates how the 2D surface changes shape, either expanding or contracting. Two types of strain were measured: 'local strain' constrained within the outline of the ablated cell and 'tissue strain' of the surrounding cells (*Figure 3F*). We found that after medial-apical ablation, anillin knockdown cells exhibited less local expansion strain compared to controls, while anillin overexpressing cells exhibited more local expansion strain compared to controls (*Figure 3F,G*). Following medial-apical ablation, there was no significant difference in tissue contraction strain between controls and anillin overexpression; however, when anillin was knocked down, the surrounding tissue exhibited less contraction strain compared to controls (*Figure 3F,G*).

Taken together, these data indicate that anillin promotes tensile force across the medial-apical surface of epithelial cells. When anillin is overexpressed, large bundles of actomyosin form, and anillin expression shifts the axis of tensile force applied on cell–cell junctions towards actomyosin bundles that exert perpendicular forces on junctions.

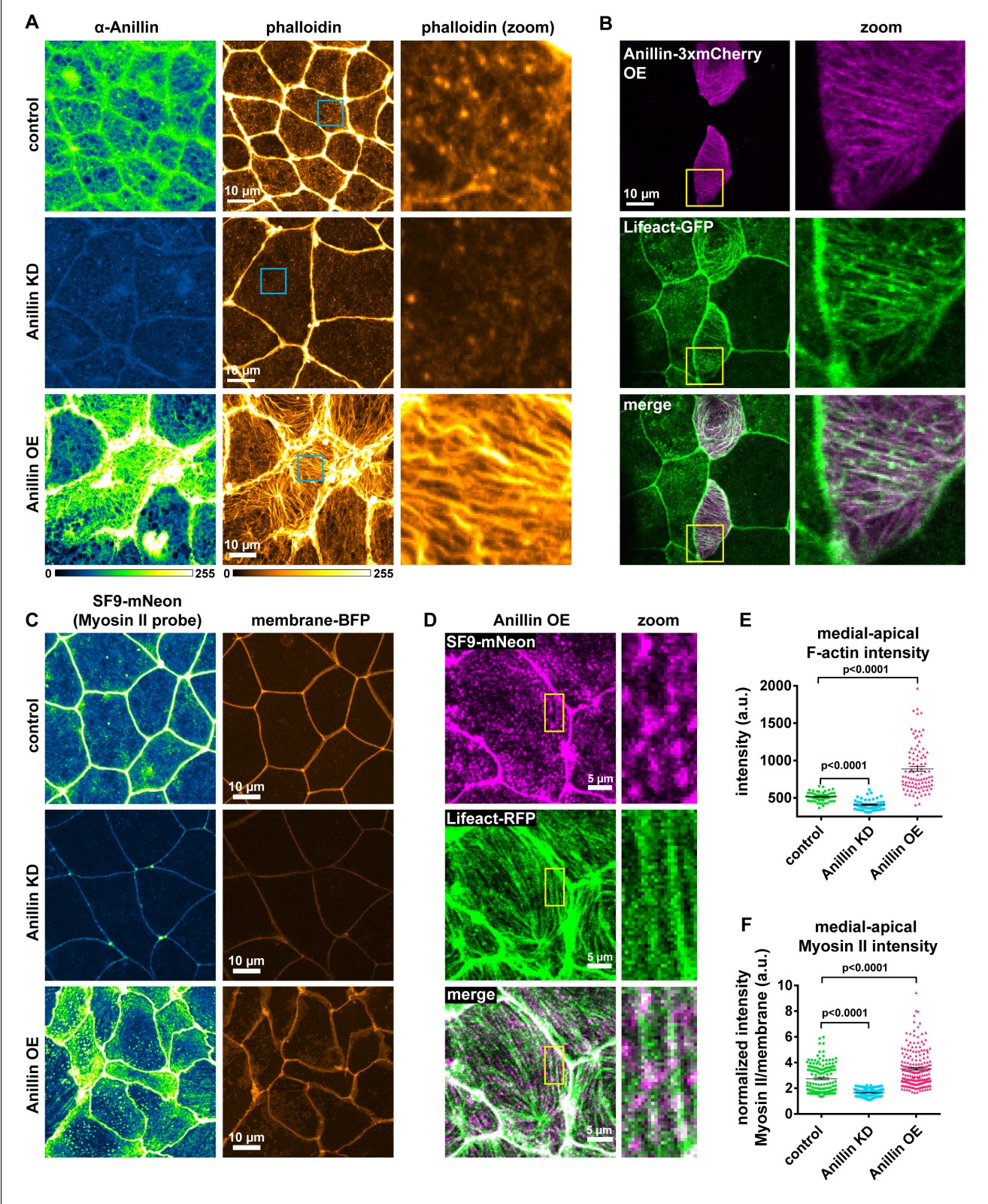

**Figure 2.** Anillin structures medial-apical F-actin and myosin II. (**A**) Confocal images of fixed epithelial cells from gastrula-stage control, anillin knockdown (KD), or anillin overexpression (OE) *Xenopus laevis* embryos stained for anillin (α-anillin) and F-actin (Alexa Fluor 488 phalloidin). Blue boxes show zoomed areas. (**B**) Confocal images of live epithelial cells expressing Anillin-3xmCherry and Lifeact-GFP (probe for F-actin). Yellow boxes show zoomed areas. (**C**) Confocal images of live epithelial cells expressing probes for myosin II (SF9-mNeon) and membrane (membrane-TagBFP). (**D**)
*Figure 2 continued on next page*

*Figure 2 continued*

Confocal images of live epithelial cells expressing probes for myosin II (SF9-mNeon) and F-actin (Lifeact-RFP). Yellow boxes show zoomed area. Notice how medial-apical myosin II decorates the F-actin bundles. (E) Quantification of medial-apical F-actin intensity from fixed embryos measured by a circular ROI 10 µm in diameter in the center of each cell. Error bars, S.E.M. Statistics, unpaired t-test. (F) Quantification of medial-apical myosin II intensity measured by a circular ROI 10 µm in diameter in the center of each cell. Medial-apical myosin II intensity was normalized by dividing by junctional membrane intensity. Error bars, S.E.M. Statistics, unpaired t-test.

DOI: https://doi.org/10.7554/eLife.39065.008

The following source data is available for figure 2:

**Source data 1.** Source data for *Figure 2E,F*.

DOI: https://doi.org/10.7554/eLife.39065.009

## Anillin establishes a contractile medial-apical actomyosin network

Our results show that when anillin is knocked down, medial-apical actomyosin is dramatically reduced and stores less contractile energy, whereas when anillin is overexpressed, actomyosin is reorganized into large bundles that store more contractile energy. Based on this we sought to determine how robust the contractile response was in these medial-apical networks. We added extracellular ATP, which leads to whole-embryo contraction driven by constriction of the apical surface of cells (*Joshi et al., 2010*; *Kim et al., 2014*; *Higashi et al., 2016*) (*Figure 4A*), making this an effective method to test the medial-apical actomyosin contractile response when anillin was perturbed. When ATP was added to one side of the animal hemisphere, control embryos contracted towards the site of ATP addition (*Figure 4A,B* and *Video 5*). The contractile response could be observed and quantified in kymographs as a shift of the pigment towards the site of ATP addition (*Figure 4B*). When anillin was knocked down, the contractile response was attenuated, and when anillin was overexpressed, the contractile response lasted longer (*Figure 4B,C*, and *Video 5*).

At the cellular level, addition of ATP to control embryos resulted in a large burst of F-actin across the medial-apical surface that dissipated over time (*Figure 4D* and *Video 6*). The burst of F-actin initially accumulated near the junctions then spread toward the center of the cell (*Figure 4—figure supplement 1A,B*, and *Video 6*). When anillin was knocked down, F-actin accumulated transient and locally at junctions, which we have previously reported as flares of active RhoA associated with local actin accumulation (*Reyes et al., 2014*; *Stephenson et al., 2019*). However, these local actin accumulations did not propagate onto the medial-apical surface and resulted in local junction contraction, but not whole-cell or whole-embryo contraction (*Figure 4D–F* and *Video 6*). When anillin was overexpressed, a burst of medial-apical F-actin similar to controls was observed, and the large bundles of F-actin thickened and became more dynamic (*Figure 4D–F* and *Video 6*). When comparing the change in medial-apical F-actin before and after ATP (baseline and peak), knocking down anillin significantly reduced the amount of F-actin accumulation compared to controls, while overexpressing anillin resulted in no significant change in the amount of F-actin accumulation (*Figure 4D,E*). Of note, medial-apical F-actin is already very high in anillin overexpressing cells compared with controls before ATP addition, so it may not be possible to achieve a significant increase in F-actin above the already elevated level.

Using another method to probe anillin's role in organizing a contractile medial-apical actomyosin network, we induced apical constriction by

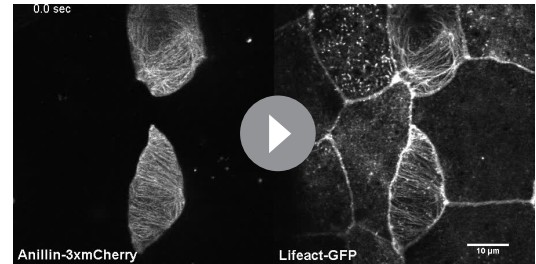

**Video 3.** Examples of medial-apical F-actin bundles caused by anillin overexpression. Cell views of a mosaic *Xenopus laevis* embryo where all cells are expressing Lifeact-GFP to label F-actin, and some cells are overexpressing Anillin-3xmCherry. Notice the cortical waves of F-actin in control cells, whereas in anillin overexpressing cells, F-actin is bundled into large filaments that span the apical surface, and filaments undergo retrograde flow from the junctions and/or rotation along the apical surface. Also note that in the lower right quadrant, there is a dividing cell that forms a contractile ring, and as the dividing cell pulls on the anillin overexpressing cell, the actin fibers orient to the direction of this pulling force.

DOI: https://doi.org/10.7554/eLife.39065.010

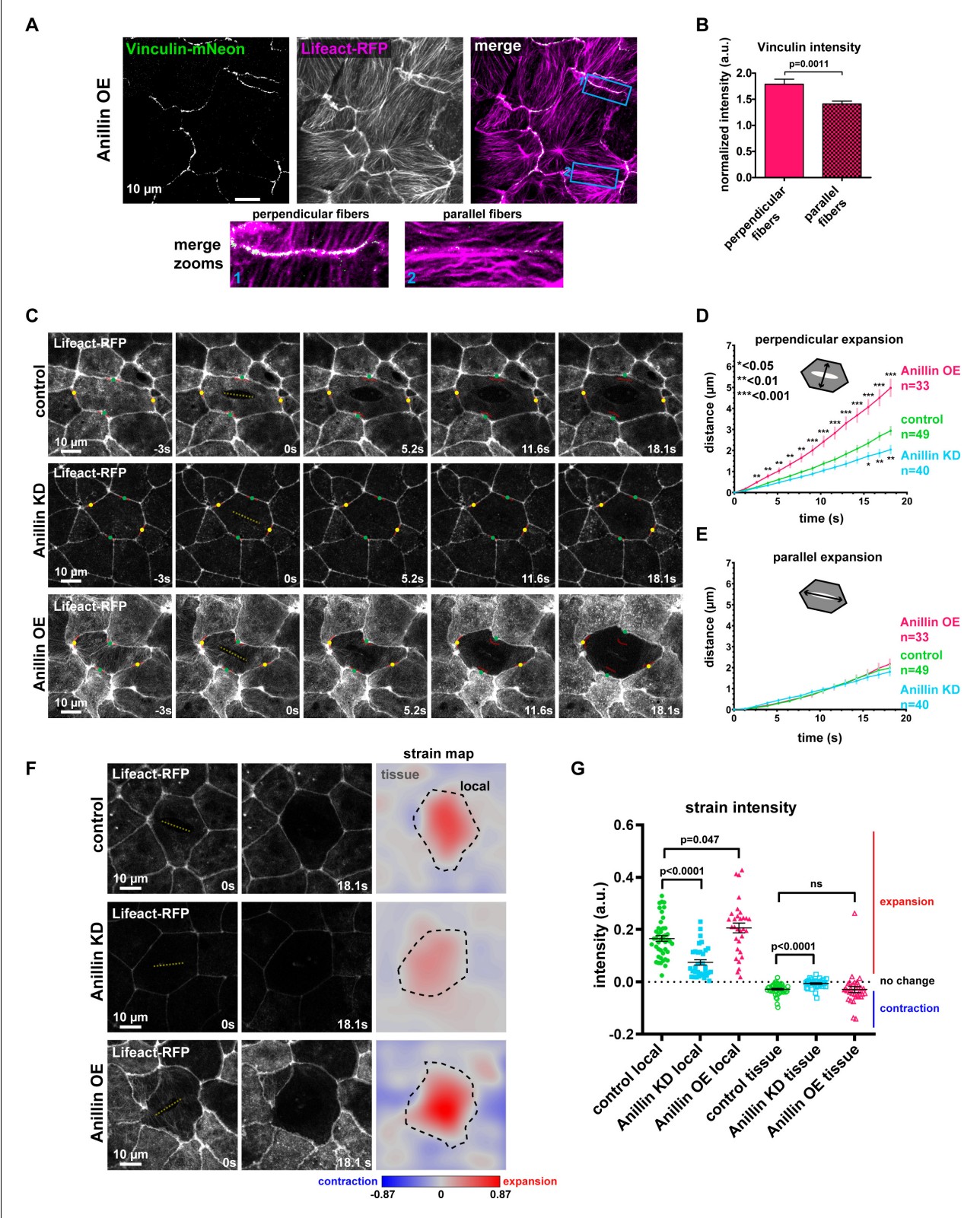

**Figure 3.** Anillin increases tensile forces and strain energy stored across the medial-apical surface of epithelial cells. (**A**) Confocal images of live epithelial cells in a gastrula-stage *Xenopus laevis* embryo expressing Vinculin-mNeon along with probes for F-actin (Lifeact-RFP) and the plasma membrane (2x membrane localization signal of Src family tyrosine kinase Lyn tagged with TagBFP at its C-terminus, not shown) when anillin is overexpressed (OE). The brightness and contrast of the Vinculin-mNeon channel was adjusted to make vinculin signal visible in the merged image;
*Figure 3 continued on next page*

*Figure 3 continued*

unadjusted images were used for quantification. (B) Quantification of vinculin intensity normalized to membrane intensity. Measurements were taken by tracing a bicellular junction from vertex to vertex at junctions with F-actin fibers connected perpendicularly to the junction or F-actin fibers running parallel to the junction. Error bars, S.E.M. Statistics, unpaired t-test, perpendicular n = 38, parallel n = 38, n = number of junctions. (C) Single-plane confocal images of medial-apical cortex ablation in cells expressing Lifeact-RFP and E-cadherin-3xGFP (not shown). Yellow dashed lines indicate the medial-apical ablation site. Red lines indicate the initial position of junctions. The points measured between junctions post-ablation are indicated by green dots (distance perpendicular to cut site) and yellow dots (distance parallel to cut site). (D) Quantification of cell opening perpendicular to the cut site measured as the junction-to-junction distance by a line drawn perpendicular to the cut site. Error bars, S.E.M. Statistics, unpaired t-test, n = number of cells. (E) Quantification of cell opening parallel to the cut site measured as the junction-to-junction distance by a line drawn parallel to the cut site. Error bars, S.E.M. Statistics, unpaired t-test, n = number of cells. (F) The first two columns show single-plane confocal images of medial-apical cortex ablation in cells expressing Lifeact-RFP at 0 s (time of ablation) and 18.1 s after ablation. The third column shows strain area mapped to the images. Pixel intensity represents area strain and indicates how the 2D surface changes shape over 18.1 s. A phase lookup table was applied to the strain maps with blue representing contraction and red indicating expansion. Yellow dashed line indicates medial-apical ablation site, black dashed line represents the 18.1 s cell outline where the area strain map intensity was measured for 'local' strain; the area outside the black dashed line was measured as 'tissue' strain. (G) Quantification of local and tissue strain mean intensities. Error bars, S.E.M. Statistics, unpaired t-test.

DOI: https://doi.org/10.7554/eLife.39065.011

The following source data and figure supplement are available for figure 3:

**Source data 1.** Source data for *Figure 3B,D,E,G* and *Figure 3—figure supplement 1A,B,C*.
DOI: https://doi.org/10.7554/eLife.39065.013

**Figure supplement 1.** Cell sizes and shapes were similar across medial-apical laser ablation experiments.
DOI: https://doi.org/10.7554/eLife.39065.012

overexpressing Shroom3, which induces apical constriction events during neurulation (*Haigo et al., 2003*; *Itoh et al., 2014*). In cells induced to apically constrict by Shroom3 overexpression, anillin accumulated across the apical surface of the cell, first near junctions then emanating towards the center of the cell (*Figure 4—figure supplement 1C*). Taken together, these data support the idea that anillin is required to establish a contractile medial-apical network that can coordinate whole-embryo contraction. When anillin is depleted, this network is not able to properly contract, disrupting cellular and embryo-wide contraction, and when anillin is overexpressed, large bundles of actomyosin enhance whole-embryo contraction. It is possible that anillin also affects the assembly rate and the physical properties of this actomyosin network, which we test later in the manuscript.

Our results indicate that anillin not only regulates junctional actomyosin (*Reyes et al., 2014*), but also organizes a medial-apical contractile actomyosin network that, when mechanically integrated across cells in an epithelial tissue (here the *Xenopus* animal cap epithelium), can affect tissue-level contraction (*Figures 1–4*). Based on our data, we speculate that anillin can regulate the orientation of tensile forces applied on cell–cell junctions as shown in the model in *Figure 5*. This model reconciles our initially surprising results that knocking down anillin leads to *decreased* junctional vinculin recruitment but *increased* recoil after laser ablation (*Figure 1A–D*). We propose that anillin knock down weakens medial-apical actomyosin to the point where it is no longer a contractile network (*Figure 5A*). This results in decreased pulling forces perpendicular to junctions, allowing the parallel forces generated by junctional actomyosin to dominate (*Figure 5B*). In contrast, overexpressing anillin results in increased medial-apical actomyosin and reorganizes the actin into thick contractile bundles that connect perpendicular to the junction (*Figure 5A*), making perpendicular forces on junctions dominant (*Figure 5B*). This working model is consistent with our data showing that when anillin is overexpressed, vinculin recruitment to junctions is increased; following junction laser ablation, vertex separation is reduced; and following medial-apical laser ablation, apical expansion is increased. Thus, our model suggests that the level of anillin expression regulates whether junctional or medial-

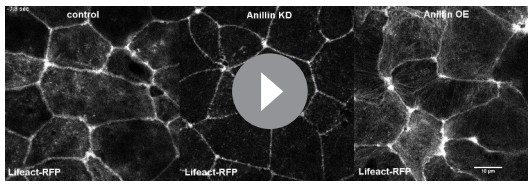

**Video 4.** Anillin increases apical expansion after medial-apical laser ablation. Cell views of the F-actin probe Lifeact-RFP in control, anillin knockdown, or anillin overexpressing *Xenopus laevis* embryos before and after medial-apical laser ablation. Ablation was performed at time 0 s.
DOI: https://doi.org/10.7554/eLife.39065.014

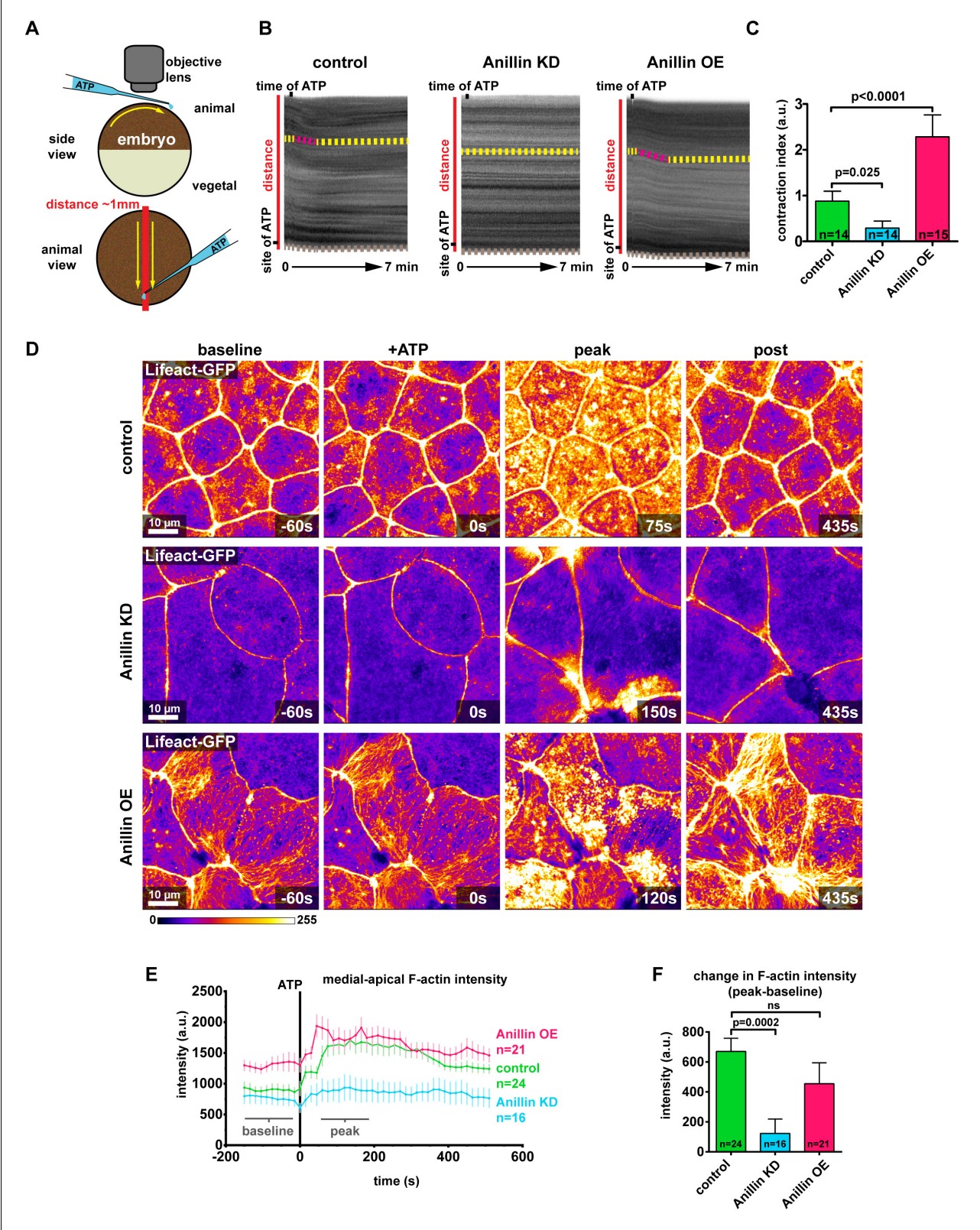

**Figure 4.** Anillin establishes a contractile medial-apical actomyosin network. (**A**) Diagram of whole embryo extracellular ATP addition experiments. 40 nl of 50 mM ATP (blue) was micropipetted onto one side of the animal hemisphere of gastrula-stage embryos. A video was captured of the embryo over time, and kymographs were generated by drawing a line across the entire embryo (red line). Yellow arrows represent the direction of contraction towards the site of ATP addition. (**B**) Kymographs of the animal hemisphere generated from light microscopy videos. The fluctuations in gray scale

*Figure 4 continued on next page*

*Figure 4 continued*

intensity are caused by the natural variations in pigment of the embryos. Contraction towards the site of ATP can be observed as a shift in the embryo's pigment towards the site of ATP addition. Yellow dashed lines represent periods of no contraction, and pink dashed lines represent contraction events. Brown dashed lines at the bottom of the kymograph represent movement of the whole embryo. (C) Quantification of the contraction index measured from kymographs. Contraction index is the difference between the shift in a pigment over time (sum of the length of yellow and pink dashed lines in B) and the movement of the whole embryo over time (brown line in B). Statistics, unpaired t-test, n = number of embryos. (D) Confocal images of live epithelial cells with F-actin probe Lifeact-GFP. After the addition of 500 µM ATP, notice the accumulation of medial-apical F-actin in controls and anillin overexpressing cells, while F-actin only increases near the junctions in anillin knockdown cells. The movement of cells across the field of view in controls and anillin knockdown was caused by flow of solution when ATP was added. (E) Quantification of medial-apical F-actin (Lifeact-GFP) intensity over time measured by a circular ROI 10 µm in diameter in the center of each cell. Statistics, unpaired t-test, n = number of cells. (F) Comparison of the change in medial-apical F-actin (Lifeact-GFP) intensity before and after ATP addition. The difference between peak and baseline F-actin intensity was measured for each embryo by averaging the peak 10 frames (from 60 s to 195 s) and subtracting the baseline 10 frames (−15 s to −150 s). Statistics, unpaired t-test, n = number of cells.

DOI: https://doi.org/10.7554/eLife.39065.015

The following source data and figure supplement are available for figure 4:

**Source data 1.** Source data for *Figure 4C,E,F* and *Figure 4—figure supplement 1B*.

DOI: https://doi.org/10.7554/eLife.39065.017

**Figure supplement 1.** In apically constricting cells, F-actin and anillin accumulation emanates from junctions and spreads medial-apically.

DOI: https://doi.org/10.7554/eLife.39065.016

apical actomyosin is the primary load-bearing structure on the apical surface of epithelial cells.

## Anillin's F-actin- and Rho-binding domains are necessary for structuring the medial-apical F-actin network

To build upon our proposed model (*Figure 5*), we examined which of anillin's N-terminal functional domains are necessary for organizing medial-apical F-actin. We made anillin mutants that lacked the myosin II-binding domain (Δmyo), the F-actin-binding domain (Δact), or both the myosin II- and F-actin-binding domains (Δmyoact) (*Figure 6A*). When overexpressed, full length anillin (Anillin FL) localized to the medial-apical surface. Overexpressed Δmyo also localized to the medial-apical surface, although less intensely; nonetheless, Δmyo reorganized F-actin into long bundles that spanned the apical surface similar to Anillin FL (*Figure 6B–D*). In contrast, overexpressed Δact localized medial-apically to a similar extent as Δmyo, but only reorganized F-actin into short bundles or no bundles at all (*Figure 6B–D*). Overexpressed Δmyoact did not efficiently localize medial-apically or reorganize F-actin robustly (*Figure 6B–D*). These results suggest that both the myosin II- and F-actin-binding domains contribute to localization of anillin to the medial-apical surface, and anillin's F-actin-binding domain is necessary for reorganizing medial-apical F-actin into long bundles.

We next examined which of anillin's C-terminal functional domains are necessary for reorganizing medial-apical F-actin. We made anillin mutants that lacked the active RhoA-binding domain (ΔRBD), the C2 domain, which binds to the plasma membrane and regulators of RhoA (ΔC2), or the PH domain, which binds the membrane, Septins, and regulators of RhoA (ΔPH) (*Figure 6E*). When overexpressed, both Anillin FL and ΔRBD were able to robustly localize to the medial-apical surface, while medial-apical localization of ΔC2 and ΔPH were reduced (*Figure 6F,G*). Overexpression of ΔRBD, ΔC2, and ΔPH did not result in structuring of medial-apical F-actin into long or short bundles, appearing similar to the negative control cells (*Figure 6F,H*). It should be noted that the ability of anillin mutants to bundle F-actin may depend on their localization to the medial-apical surface. If ΔC2 and ΔPH localized to the medial-apical surface at similar levels to Anillin FL, it is possible that they might be able

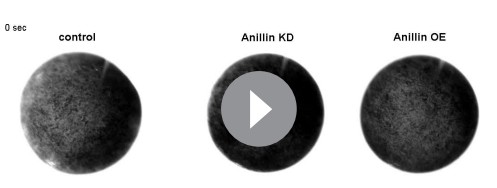

**Video 5.** Anillin promotes whole embryo contraction after addition of exogenous ATP. Whole *Xenopus laevis* embryo views where ATP is added to the bottom center of embryos while live imaging. The thick part of the microinjection needle that dispenses ATP can be seen at the top right of the embryo: these areas were avoided when making the kymographs in *Figure 4*. Each embryo is ~1.2 mm in diameter.

DOI: https://doi.org/10.7554/eLife.39065.018

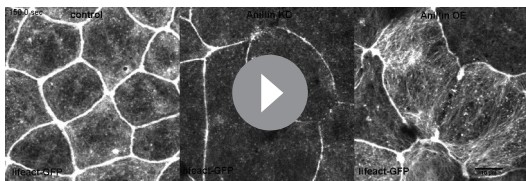

**Video 6.** Anillin promotes accumulation of medial-apical F-actin after addition of exogenous ATP. Cell views of Lifeact-GFP to label F-actin in control, anillin knockdown, or anillin overexpressing *Xenopus laevis* embryos where ATP was added to the imaging chamber while live imaging. ATP was added at time 0 s.

DOI: https://doi.org/10.7554/eLife.39065.019

to bundle F-actin. Together, these results indicate that anillin's F-actin-binding and Rho-binding domains are necessary for structuring medial-apical F-actin into long bundles that span the apical surface, and anillin's medial-apical localization is facilitated by actomyosin binding with possible contributions from binding to the plasma membrane, regulators of RhoA, or Septins.

## Anillin regulates epithelial cell mechanics by stabilizing medial-apical F-actin

Our findings demonstrated that anillin's actin-binding function is necessary to reorganize medial-apical F-actin, and previous work has shown that anillin stabilizes F-actin by preventing Cofilin severing of actin filaments (*Tian et al., 2015*). This led us to ask whether anillin stabilizes medial-apical F-actin, and if actin stabilization contributes to changes in epithelial cell mechanics. To test whether anillin stabilizes F-actin, we used Fluorescence Recovery After Photobleaching (FRAP) of Actin-mNeon in controls cells and in cells overexpressing anillin (*Figure 7A–C*). After photobleaching in control cells, Actin-mNeon recovered to 77 ± 2%, whereas in cells overexpressing anillin, Actin-mNeon recovered to 57 ± 4% (*Figure 7A–C*), indicating that anillin stabilizes actin filaments. We found that anillin requires its actin-binding domain to stabilize medial-apical F-actin; when the Anillin Δact mutant was overexpressed, F-actin recovered to a similar level as controls, 78 ± 4%. We performed FRAP of Actin-mNeon when anillin was knocked down, but because anillin knockdown drastically reduces medial-apical F-actin, we had to inject excessive amounts of Actin-mNeon mRNA in order to even visualize medial-apical F-actin. This high level of Actin-mNeon led to abnormal cell and embryo morphology in control and anillin overexpression embryos. In anillin knockdown cells, the overexpressed Actin-mNeon recovered to 96 ± 8%; however, these data cannot be directly compared with control and anillin overexpression data because of the differing levels of Actin-mNeon expression (*Figure 7—figure supplement 1A*).

Because the FRAP data revealed that anillin not only reorganizes F-actin, but also stabilizes actin filament dynamics, we hypothesized that stabilizing F-actin using the drug jasplakinolide (*Holzinger, 2001*) could produce changes in cell mechanics similar to anillin overexpression. After treating embryos with jasplakinolide (20 µM, 1 hr), we observed increased F-actin bundles at the medial-apical surface (*Figure 7D* and *Video 7*). These actin bundles were shorter than those observed when anillin was overexpressed, didn't span the apical surface, and didn't flow towards the center of the cell (*Figure 7D*, *Video 7*, and *Video 3*). We performed junctional laser ablation to test whether the effect observed when F-actin was stabilized by jasplakinolide would recapitulate that of anillin overexpression (reduced parallel recoil and instances of dramatic perpendicular recoil). Indeed, jasplakinolide treatment strongly reduced parallel recoil, from 2.7 ± 0.3 µm after 18 s in controls to 0.2 ± 0.1 µm after 18 s (*Figure 7E,F*, and *Video 8*). Furthermore, jasplakinolide treatment of anillin knockdown embryos also reduced junction recoil after laser ablation (*Figure 7—figure supplement 1B*). Jasplakinolide treatment did not increase the number of cells that separate perpendicularly compared to controls, it did, however, rescue perpendicular separation rates in anillin knockdown cells (*Figure 7—figure supplement 1C*). Remarkably, following jasplakinolide treatment of control embryos, we observed several examples of dramatic perpendicular recoil as well as junctional splaying of E-cadherin-3xGFP, similar to the effect when anillin was overexpressed (*Figure 7G* and *Figure 1F*). In some cases, for both jasplakinolide treated cells and anillin-overexpressing cells, we observed that parallel vertex separation only occurred after perpendicular separation near the vertex occurred (*Figure 7G*, *Figure 7—figure supplement 1E–G*, and *Video 8*), indicating that forces perpendicular to these junctions were maintaining cell shape after laser ablation. Together, these results demonstrate that anillin stabilizes medial-apical F-actin and suggest that F-actin stability contributes to the mechanical changes observed in epithelial cells when anillin is perturbed.

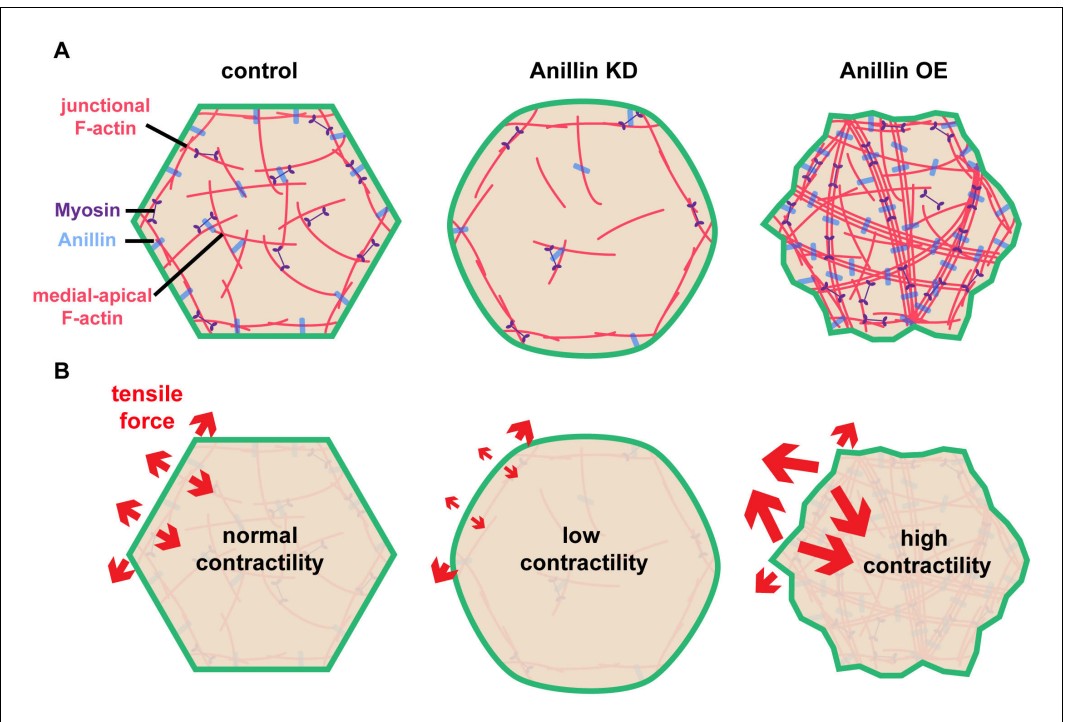

**Figure 5.** Anillin organizes medial-apical actomyosin and regulates the orientation of tensile forces applied on junctions. (**A**) Diagram of the apical surface of epithelial cells showing junctional and medial-apical F-actin and anillin in control embryos or when levels of anillin are perturbed. When anillin is knocked down, medial-apical F-actin is strongly reduced, and when anillin is overexpressed, medial-apical F-actin is reorganized into thick bundles decorated with myosin II that span the apical surface. (**B**) Diagram depicting the changes in the orientation of actomyosin-mediated forces applied on cell–cell junctions when anillin is perturbed. When anillin is knocked down, the medial-apical actomyosin is not robustly contractile, reducing forces perpendicular to the junction. When anillin is overexpressed, perpendicular forces on cell–cell junctions are increased due to the large contractile bundles of F-actin that connect perpendicular to the junction.

DOI: https://doi.org/10.7554/eLife.39065.020

## Anillin promotes tissue stiffness

Since anillin promotes medial-apical actin stability and reduces junction recoil after laser ablation, we next asked whether anillin promotes tissue-wide stiffness. To measure the effect that anillin has on tissue stiffness, we utilized dorsal isolates from neurula-stage (Nieuwkoop and Faber stage 15) *Xenopus laevis* embryos, as they are amenable to measuring tissue stiffness (*Figure 8A*) (*Zhou et al., 2009*). First, we characterized the effects of anillin knockdown on F-actin in the dorsal isolates by knocking down anillin in half of the embryo, leaving the other half of the embryo as an internal control. Similar to our results in the gastrula-stage animal cap epithelium (*Figure 2A,E*), regions of the dorsal isolates where anillin was knocked down showed a marked reduction in junctional and medial-apical F-actin compared with internal control regions (*Figure 8B,C*, and *Figure 8—figure supplement 1*). Interestingly, transverse sections of dorsal isolates revealed that anillin knockdown not only reduced apical F-actin but also enhanced accumulation of F-actin at the basal surface of these epithelial cells in the ectoderm (*Figure 8C* and *Figure 8—figure supplement 1*). Anillin knockdown also disrupted cell morphology causing doming in both ectodermal and endodermal cells, similar to our previous report in gastrula-stage epithelia (*Figure 8C* and *Figure 8—figure supplement 1*) (*Reyes et al., 2014*). In mesodermal cells, anillin knockdown disrupted the stereotypic patterning, leading to elongated cell size and accumulation of F-actin at cellular poles along the long axis of the cells (*Figure 8C* and *Figure 8—figure supplement 1*). When anillin knockdown cells were found in the notochord, they exhibited irregular shapes, abnormal F-actin accumulation, disrupted the stereotypic cross-sectional ellipsoid shape of the notochord itself, and disrupted tissue interfaces between notochord, somite and medial endoderm (*Figure 8C* and *Figure 8—figure supplement 1*).

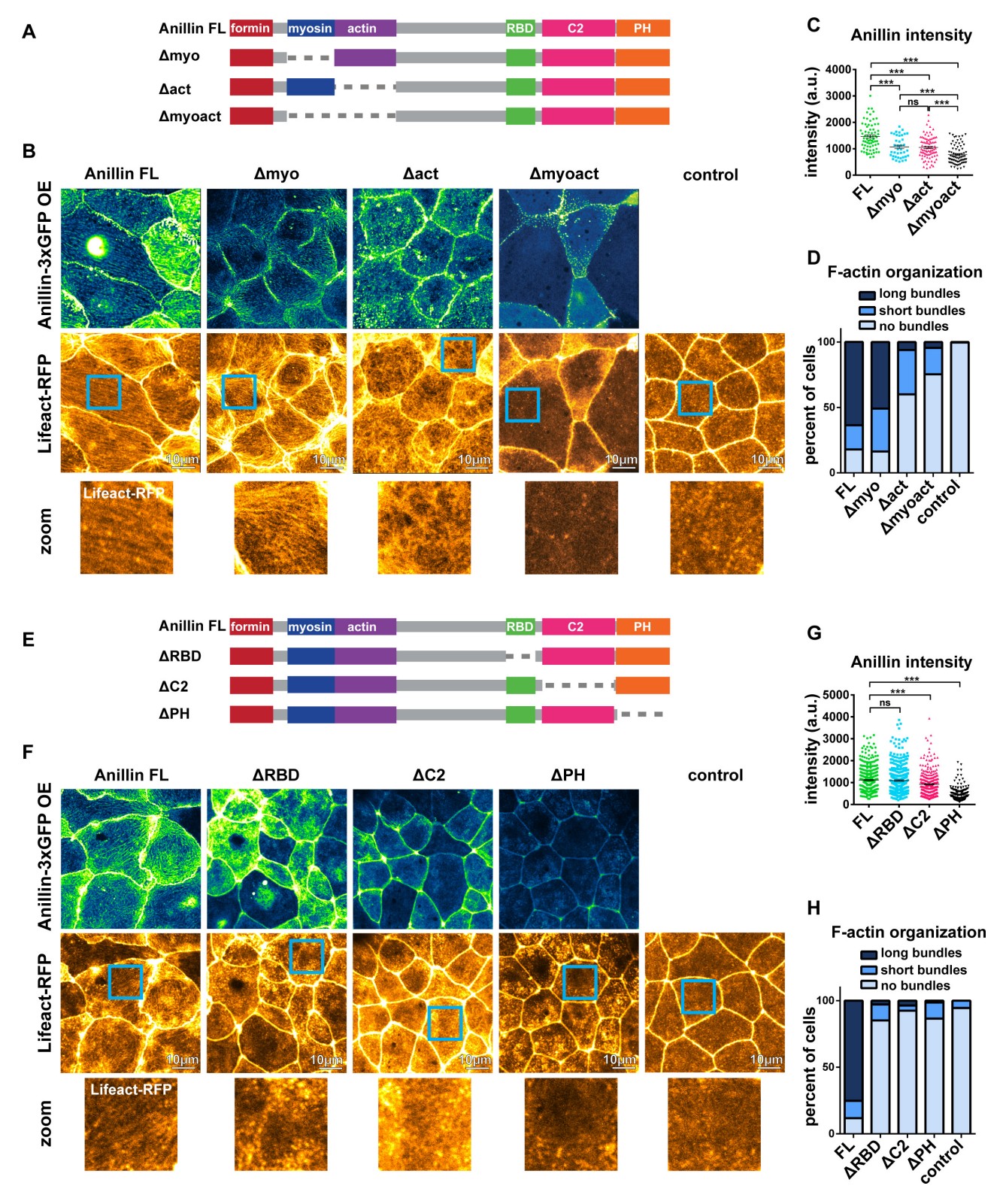

**Figure 6.** Anillin's F-actin- and Rho-binding domains are necessary for structuring the medial-apical F-actin network. (**A**) Domain diagram of full length anillin and N-terminal domain deletions. Full length (FL), Rho binding domain (RBD), pleckstrin homology (PH). (**B**) Confocal images of live epithelial cells from gastrula-stage *Xenopus laevis* embryos expressing either Anillin FL or anillin domain deletions tagged with 3xGFP along with Lifeact-RFP to probe for F-actin. Blue boxes show zoomed areas. (**C**) Quantification of anillin construct intensity measured by a circular ROI 10 μm in diameter in the

*Figure 6 continued on next page*

*Figure 6 continued*

center of each cell. Statistics, unpaired t-test, *** p =<0.001, n = number of cells. (D) Blinded categorization of medial-apical F-actin bundle organization in cells. FL, n = 306; Δmyo, n = 112; Δact, n = 61; Δmyoact, n = 57; control, n = 110 cells. (E) Domain diagram of Anillin FL and C-terminal domain deletions. (F) Confocal images of live epithelial cells from gastrula-stage *Xenopus laevis* embryos expressing either Anillin FL or anillin domain deletions tagged with 3xGFP and Lifeact-RFP to probe for F-actin. (G) Quantification of anillin construct intensity measured by a circular ROI 10 µm in diameter in the center of each cell. Statistics, unpaired t-test, *** p =<0.001, n = number of cells. (H) Blinded categorization of medial-apical F-actin bundles in cells. FL, n = 395; ΔRBD, n = 352; ΔC2, n = 332; ΔPH, n = 287; control, n = 109 cells.

DOI: https://doi.org/10.7554/eLife.39065.021
The following source data is available for figure 6:

**Source data 1.** Source data for *Figure 6C,D,G,H*.
DOI: https://doi.org/10.7554/eLife.39065.022

To test whether anillin promotes tissue stiffness, anillin was knocked down throughout the embryo. Despite delayed gastrulation, the anillin knockdown embryos were able to develop to stage 15, and dorsal isolates were loaded onto a nanoNewton Force Measurement Device, which measures resistive forces generated by the dorsal isolate in response to compression (*Figure 8D*) (*Zhou et al., 2009*). Stiffness (defined as the elastic modulus after 180 s of unconfined compression) is calculated using measured resistive force, compressive strain, and the dorsal isolate cross-sectional area (*Zhou et al., 2009*). There was a significant reduction in the stiffness of dorsal isolates when anillin was knocked down compared with controls (*Figure 8E*). Taken together, these results indicate that anillin promotes tissue stiffness, which may be due to anillin's role in organizing medial-apical F-actin. It should be noted, however, that the loss of junctional F-actin or the defects in presomitic mesoderm, notochord, or endoderm may also contribute to the measured change in tissue stiffness.

## Discussion

Anillin is a scaffolding protein with diverse binding capabilities, which it uses to organize the cytoskeletal elements including F-actin, Septins, and microtubules (*Field and Alberts, 1995*; *Sisson et al., 2000*; *Silverman-Gavrila et al., 2008*; *D'Avino, 2009*), build actin structures by activating formins (*Watanabe et al., 2010*), and promote contractility by linking F-actin and myosin II to the plasma membrane and binding active RhoA and its regulators (*Straight et al., 2005*; *Piekny and Glotzer, 2008*; *Sun et al., 2015*; *Manukyan et al., 2015*; *Descovich et al., 2018*). With this diverse set of interactions, it is not surprising that anillin functions in a range of cellular processes across the tree of life. It is required for cytokinesis in many organisms, from yeast to vertebrates, promotes cellularization in fly embryos, maintains bridges between germ cells in worms, and regulates migration in neurons (*Field and Alberts, 1995*; *Piekny and Maddox, 2010*; *Amini et al., 2015*; *Sun et al., 2015*). Furthermore, anillin has an emerging role in regulating cell–cell junctions and promoting proper tissue barriers through maintaining the contractile actomyosin network connected to cell–cell junctions (*Gbadegesin et al., 2014*; *Reyes et al., 2014*; *Wang et al., 2015*; *Budnar et al., 2018*). Here, we have discovered a new role for anillin in organizing a contractile network of medial-apical actomyosin in frog embryonic epithelia. We demonstrate that perturbing anillin changes the mechanical properties of both individual cells and the tissue as a whole (*Figure 9*).

### Multiple factors contribute to junction recoil after laser ablation

Laser ablation has been a useful tool for inferring the relative amount of tension on cell–cell junctions, leading to many insights about the proteins involved in generating tension on junctions (*Farhadifar et al., 2007*; *Fernandez-Gonzalez et al., 2009*; *Ratheesh et al., 2012*; *Leerberg et al., 2014*; *Van Itallie et al., 2015*; *Choi et al., 2016*; *Priya et al., 2017*; *Bertocchi et al., 2017*). Generally, junctional laser ablation is thought to measure the amount of line tension or tension in the direction parallel to the edge of a cell (*Rauzi and Lenne, 2015*). However, there is also evidence that the medial-apical actin network can serve as a load-bearing structure within the cell (*Ma et al., 2009*).

Our junctional laser ablation experiments produced unexpected results with respect to the prevalent line tension paradigm (*Figure 9A*) and led us to characterize a new function for anillin in organizing contractile medial-apical actomyosin structures. Junctional laser ablation indicated that anillin knockdown cells exhibited increased recoil, whereas anillin overexpressing cells exhibited reduced

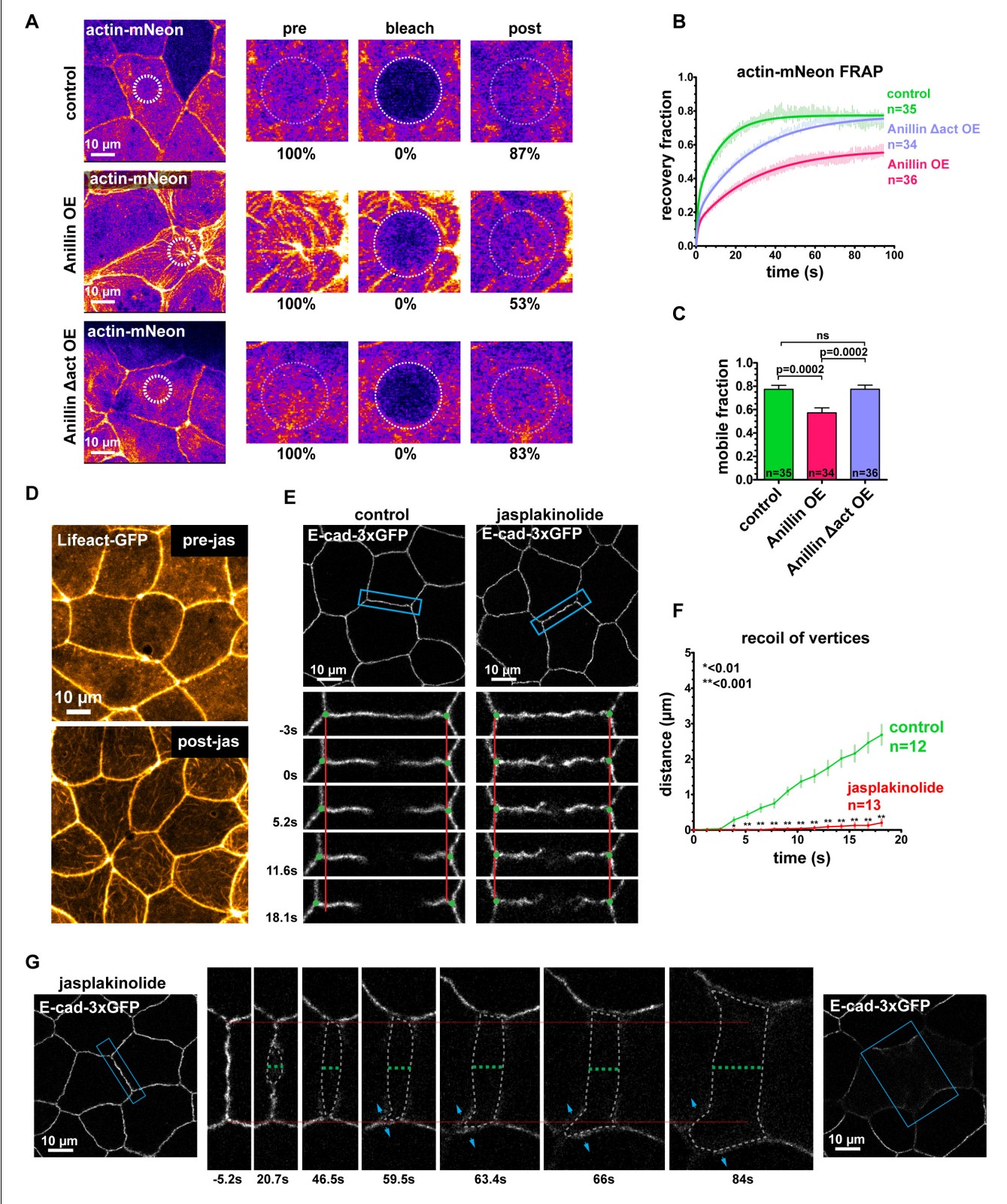

**Figure 7.** Anillin regulates epithelial cell mechanics by stabilizing medial-apical F-actin. (**A**) Single plane confocal images of live epithelial cells from gastrula-stage *Xenopus laevis* embryos expressing actin tagged with mNeon (Actin-mNeon) in control, anillin overexpressing (OE), or Anillin Δactin-binding domain OE embryos. White dashed circle shows the medial-apical area bleached in the zoomed view montages on the right. The percent of original fluorescence intensity (normalized to 100%) is indicated below each of the zoomed views. (**B**) Medial-apical Actin-mNeon Fluorescence

*Figure 7 continued on next page*

*Figure 7 continued*
Recovery After Photobleaching (FRAP) data fitted with a two phase association curve. n = number of cells. (C) Mobile fractions from FRAP data. Error bars, S.E.M. Statistics, unpaired t-test, n = number of cells. (D) Confocal images of live epithelial cells expressing a probe for F-actin (Lifeact-GFP). Top image was taken before the addition of 20 µm jasplakinolide. Bottom image of the same field of view was taken 1 hr after the addition of jasplakinolide. (E) Single plane confocal images showing E-cadherin tagged with 3xGFP (E-cad-3xGFP) prior to junction laser ablation for a control embryo (2% EtOH) and an embryo treated with 20 µm jasplakinolide. Blue boxes show the zoomed area for the ablation montage. Red lines indicate the initial location of junction vertices, green dots indicate the location of junction vertices measured after ablation. (F) Quantification of vertex separation over time after ablation (at time 0). Error bars, S.E.M. Statistics, unpaired t-test, n = number of cells (G) Confocal images of an embryo expressing E-cad-3xGFP and treated with 20 µm jasplakinolide before and after laser ablation. Blue boxes show the zoomed area for the ablation montage. Red lines indicate the initial position of the vertices, green dashed line indicates the perpendicular separation between the two cells, gray dashed line indicates the space forming between the two cells, and blue arrows represent forces on junctions adjacent to the lower cell vertex. Notice that the lower vertex only begins to separate in the parallel direction after the forces perpendicular to the adjacent junction (blue arrows) lead to loss of adhesion between the two cells.
DOI: https://doi.org/10.7554/eLife.39065.023

The following source data and figure supplement are available for figure 7:

**Source data 1.** Source data for *Figure 7B,C,F* and *Figure 7—figure supplement 1A,B,C*.
DOI: https://doi.org/10.7554/eLife.39065.025

**Figure supplement 1.** Stabilizing F-actin rescues anillin knockdown junction recoil defect after laser ablation.
DOI: https://doi.org/10.7554/eLife.39065.024

recoil compared with controls. These results were unexpected because all other evidence (vinculin recruitment) (*Figure 1A,B*), F-actin and myosin II junctional intensity (*Reyes et al., 2014*), cell shape changes (*Reyes et al., 2014*), suggested that junctional tension was reduced when anillin was knocked down and increased when anillin was overexpressed. If we had performed either the vinculin recruitment or junctional laser ablation experiments in isolation, opposite conclusions would have been drawn about how the anillin expression level affects junctional tension. A weakness of these tension probing techniques is that they do not easily reveal the direction of forces acting on the junction. For example, if a junction recoils slowly parallel to the junction after laser ablation, is this because there is less parallel tension, more perpendicular tension, or other mechanical changes? By performing medial-apical laser ablation (*Figure 3C–G*), we were able to show that anillin promotes a medial-apical actomyosin network that produces tensile force and stores contractile energy that acts perpendicular to cell–cell junctions (*Figure 9B*).

Through FRAP studies, we found that anillin stabilizes medial-apical F-actin, which may be important for establishing this network as a load-bearing structure (*Figure 9C*). By stabilizing F-actin with jasplakinolide, we found that junction recoil was dramatically decreased, similar to the effect with anillin overexpression, indicating that cellular mechanics are also regulated by the physical properties of the F-actin network (*Figure 9C*) in addition to the contractile forces generated by the network.

Finally, we showed that anillin promotes tissue-wide stiffness, and this likely contributes to the changes we observe in cellular mechanics, as well (*Figure 9D*). When anillin was knocked down, the stiffness of *Xenopus* dorsal isolates was reduced, or put in other terms, the tissue compliance was increased. While there is little direct evidence about how tissue compliance affects laser ablation results, it is easy to conceive how two junctions under the same tension - one in a compliant tissue vs. one in a non-

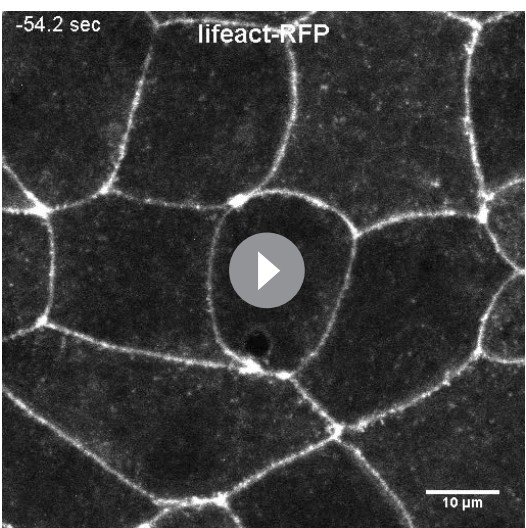

**Video 7.** Stabilizing F-actin with jasplakinolide produces medial-apical actin bundles similar to when anillin is overexpressed. Cell views of Lifeact-RFP used to label F-actin in a *Xenopus laevis* embryo treated with Jasplakinolide. Jasplakinolide was added at time 0 s.
DOI: https://doi.org/10.7554/eLife.39065.026

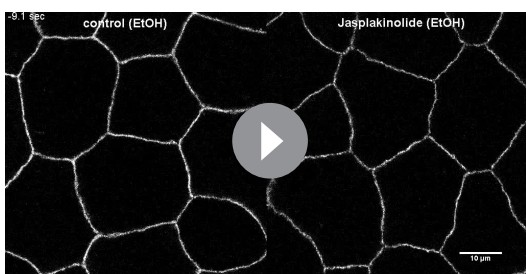

**Video 8.** Stabilizing F-actin with jasplakinolide reduces vertex separation following junctional laser ablation similar to when anillin is overexpressed. Cell views of the F-actin probe Lifeact-RFP in control *Xenopus laevis* embryo (treated with vehicle, EtOH) or embryo treated with Jasplakinolide before and after junctional laser ablation. Ablation was performed at time 0 s.
DOI: https://doi.org/10.7554/eLife.39065.027

compliant tissue - would recoil at different velocities. A recent study found that cell boundary stiffness was locally increased by unidirectional actomyosin-generated tension, and when ablated parallel to this axis, the tissue displayed high recoil (*Chanet et al., 2017*). In contrast, when actomyosin-generated contractility was omnidirectional, cell boundary stiffness increased evenly around the cell boundary, and when ablated, the tissue displayed reduced recoil (*Chanet et al., 2017*). While changes in the axis of actomyosin contractility certainly contribute to the changes in junction recoil, the changes in boundary stiffness may also play an essential role. In the future, it would be interesting to make more robust stiffness measurements at the apical surface using techniques such as atomic force microscopy or micro-aspiration. Our results emphasize the emerging point that studies using laser ablation or other methods of measuring junctional tension should consider potential contributions of perpendicular tensile forces generated by the medial-apical actomyosin network, the stability of F-actin filaments, and the compliance of the junction and surrounding tissue.

## Anillin regulates apical contractility in epithelia

Combining our previous study, which showed that anillin maintains F-actin, myosin II, and proper RhoA activity at cell–cell junctions (*Reyes et al., 2014*), with this study, which shows that anillin promotes a contractile medial-apical network, positions anillin as a key regulator of contractility at the apical surface of epithelial cells. The medial-apical population of actomyosin has previously been characterized, particularly in flies, and is critical for constricting the apical surface of epithelial cells to drive tissue bending during embryonic development (*Dawes-Hoang et al., 2005*; *Martin et al., 2009*; *Plageman et al., 2011*; *Martin and Goldstein, 2014*). Our results suggest that the level of anillin expression may modulate redistribution of cellular load-bearing structures from junctional/circumferential when anillin is expressed at a low level to cortex-spanning/medial-apical when anillin is expressed at a high level. This shift parallels one seen during *Drosophila* development over the course of germ band elongation. Initially, tissue tension is transmitted primarily through junctional structures, but by the end of germ band elongation, it is carried by cortex-spanning structures (*Ma et al., 2009*). Additional work will be necessary to distinguish the functions of anillin-organized contractility at junctional and medial-apical contractile arrays.

Anillin's role in regulating medial-apical contractility may function to prevent junction disassembly by generating forces perpendicular to the junction. myosin II-generated tension applied on adherens junctions can prevent junction disassembly (*Weng and Wieschaus, 2016*). Here, we showed that anillin-organized medial-apical contractility produces perpendicular tension on adherens junctions. This raises the possibility that protective tensile forces organized by medial-apical anillin are lost when anillin is knocked down, which could lead to the compromised junctions we have previously reported in anillin knockdown cells (*Reyes et al., 2014*).

Our findings establish anillin as a candidate protein that may be involved in developmental events that require apical constriction including gastrulation and neurulation. Anillin's potential role in development is supported by the fact that many of anillin's interacting proteins - including active RhoA, the formin Diaphanous, F-actin, and myosin II - also localize to the medial-apical actomyosin network and are required for apical constriction events during development (*Mason et al., 2013*; *Martin and Goldstein, 2014*). Of note, when Shroom3, which is known to induce apical construction during neurulation (*Haigo et al., 2003*), was overexpressed, anillin strongly accumulated across the medial-apical surface as apical constriction proceeded (*Figure 4—figure supplement 1C*). In the future, it will be interesting to directly test whether anillin plays a role in driving apical constriction during gastrulation and/or neurulation.

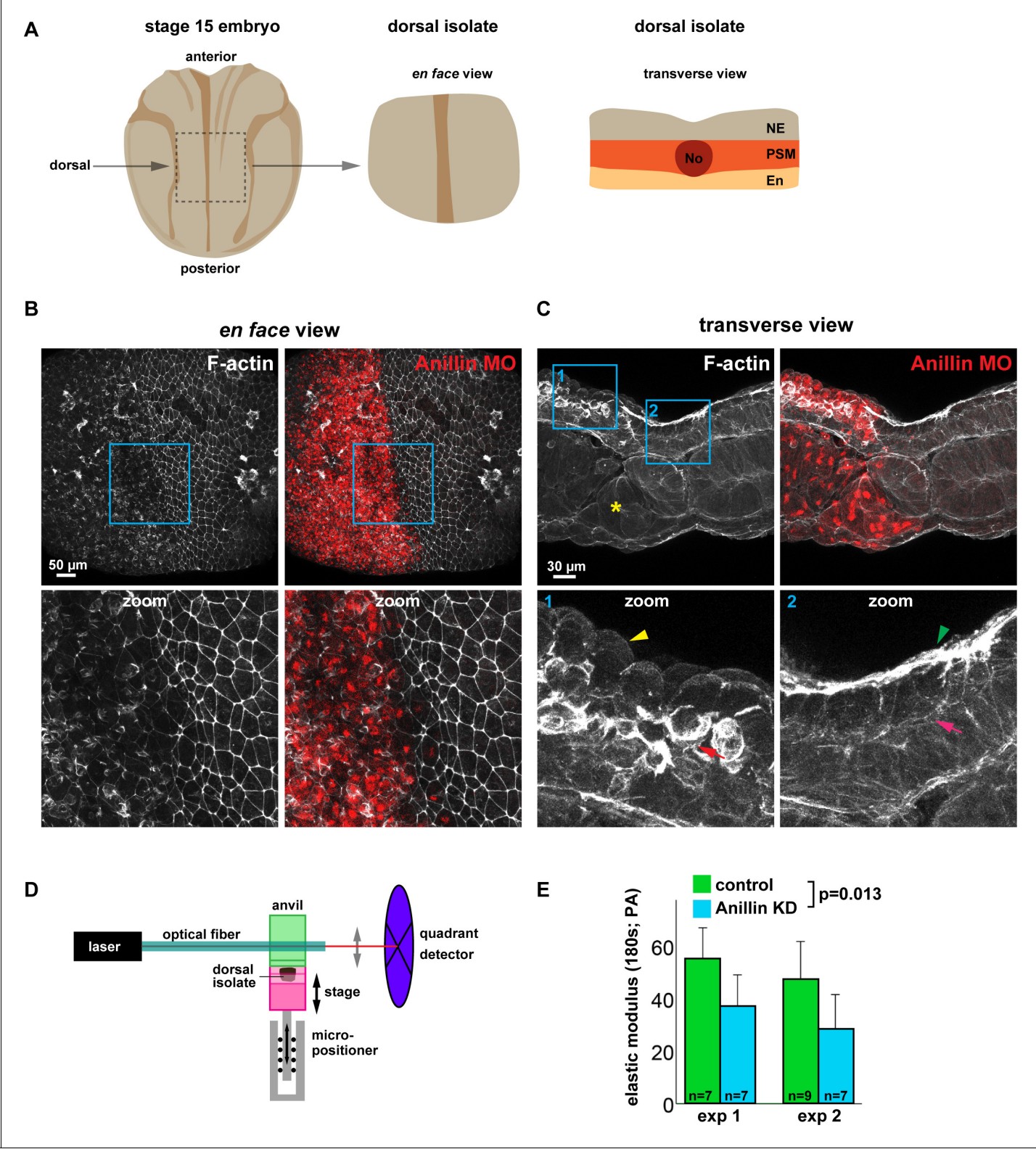

**Figure 8.** Anillin promotes tissue stiffness. (**A**) Cartoons of a stage 15 *Xenopus laevis* embryo and dorsal isolate (based on Nieuwkoop and Faber). Transverse view shows the neural ectoderm (NE), presomitic mesoderm (PSM), endoderm (En), and notochord (No). (**B**) *En face* images of fixed dorsal isolates, z-projected (~50 microns deep). F-actin was stained using phallacidin tagged with BODIPY FL. Anillin morpholino was co-injected with Alexa Fluor 647-conjugated Dextran as a lineage tracer. Blue boxes represent zoomed areas below. Notice the reduced junctional and medial-apical F-actin in anillin-depleted neural ectoderm. (**C**) Transverse section views of the dorsal isolates. Blue boxes show zoomed areas below. Anillin

*Figure 8 continued on next page*

*Figure 8 continued*

knockdown epithelial cells exhibit an apically domed morphology and loss of F-actin on their apical surface (yellow arrowhead), enhanced basolateral F-actin (red arrow), and disrupted mesoderm organization (yellow asterisk). Control epithelial cells have a flat apical morphology with robust F-actin (green arrowhead) and little basal F-actin (pink arrow). (D) Diagram of the nanoNewton Force Measurement Device, which measures resistive force generated by the tissue isolate in response to compression using a force-calibrated optical fiber. A computer-controlled stage compresses the tissue isolate against an anvil attached to the optical fiber, and the resistive force is measured using the deflection of the optical force probe. (E) Quantification of the stiffness of dorsal isolates. Tissue stiffness is significantly reduced when anillin is knocked down. Statistics, 2-way ANOVA, n = number of dorsal isolates.

DOI: https://doi.org/10.7554/eLife.39065.028

The following source data and figure supplement are available for figure 8:

**Source data 1.** (*Figure 8E*) Dorsal isolate elastic modulus with anillin knockdown.
DOI: https://doi.org/10.7554/eLife.39065.030
**Figure supplement 1.** Anillin maintains apical F-actin in dorsal epithelium.
DOI: https://doi.org/10.7554/eLife.39065.029

## Does anillin regulate tissue stiffness in development and disease?

Our data demonstrating that anillin promotes tissue stiffness is intriguing because tissue stiffness can affect tissue folding during development (*Jackson et al., 2017*) as well as cell migration during development and in cancer (*Barriga et al., 2018*). Although we cannot confirm whether reduction in tissue stiffness after anillin knockdown is directly related to loss of medial-apical anillin vs. contributions from loss of junctional anillin, failed cell division events, or the changes in other tissue layers, the idea that anillin is regulating tissue stiffness by organizing actomyosin contractility at one or more of these sites is supported by previous work. For example, tissue stiffness increases during *Xenopus* development, and this is dependent on F-actin and myosin II (*Zhou et al., 2009*). Indeed, the ~35% reduction in stiffness we observed after anillin knockdown is in line with the ~50% reduction in stiffness observed after acute drug treatments that disrupt F-actin or myosin II (*Zhou et al., 2009*). Additionally, a recent study directly connected medial-apical contractility with promoting tissue stiffness, and found that the direction of contractility promoted oriented tissue stiffness, which in turn drove proper tissue invagination (*Chanet et al., 2017*). Future studies should investigate whether anillin impacts developmental events that require tissue stiffening.

Our findings may also provide insight into the role anillin plays in cancer progression. To date, studies examining anillin's role in tumor progression and survival outcome have produced conflicting evidence. Most studies have found that anillin is overexpressed in diverse tumors, and its expression correlates with cancer progression and poor survival rates (*Hall et al., 2005*; *Suzuki et al., 2005*; *Wang et al., 2016*; *Idichi et al., 2017*; *Zhang et al., 2018*). However, work examining the subcellular localization of anillin found that *nuclear* localization of anillin correlates with poor survival rates, whereas *cytoplasmic* localization of anillin is a marker of favorable prognosis, suggesting that anillin's localization rather than its expression level is key (*Ronkainen et al., 2011*; *Liang et al., 2015*). Our finding that anillin increases tissue stiffness may help explain anillin's role in tumor progression. It is well known that tumors increase in stiffness over time due to increased extracellular collagen, and this stiffening promotes tumor progression (*Fang et al., 2014*); however, measurements of the stiffness of individual cancer cells revealed that cells become less stiff as they become metastatic (*Swaminathan et al., 2011*; *Guo et al., 2014a*). Considering our results in this context, we speculate that *cytoplasmic* anillin could help protect against cancer progression, as anillin stiffens the cells and the tissue by reorganizing actomyosin, preventing cells from entering a migratory metastatic state. Our previous work showing that anillin is important for maintaining normal cell–cell adhesion could also play a role in preventing cancer cell invasion and metastasis (*Reyes et al., 2014*).

## Conclusions

Our results reveal that by structuring medial-apical actomyosin, anillin helps orchestrate apical contractility in epithelial cells, affects the orientation of tensile forces applied on junctions, and promotes tissue stiffness. Thus, anillin is potentially a key cytoskeletal organizer in events that require apical constriction, junction maintenance, and/or changes in tissue stiffness. Future work is required to tease apart the contributions and interplay among anillin's versatile roles in cytokinesis, cell–cell

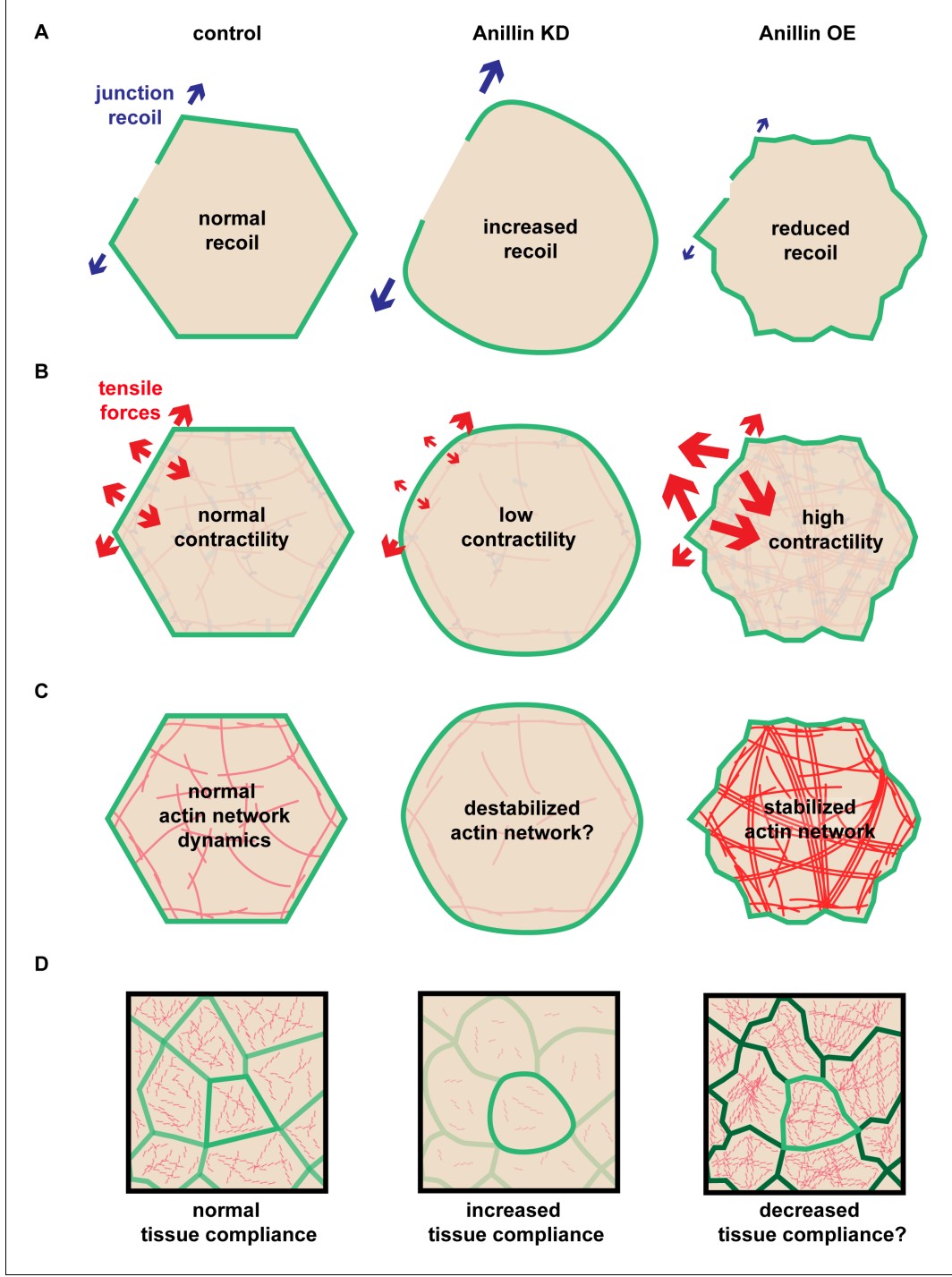

**Figure 9.** Anillin regulates apical tensile forces, stabilizes F-actin, and stiffens tissues. (**A**) Schematic of our results showing that increased anillin expression slows junction recoil. (**B**) Diagram of how anillin slows junction recoil by reorienting tensile forces across the apical surface of cells, changing the orientation of the dominant tensile force from being in line with the junction to being perpendicular to the junction, which may contribute to the observed changes in junction recoil. (**C–D**) Diagram of how anillin stabilizes F-actin (**C**) and modulates tissue compliance (**D**), both of which may contribute to the observed changes in junction recoil.

DOI: https://doi.org/10.7554/eLife.39065.031

adhesion, and medial-apical actomyosin regulation and how they culminate to promote a functional

epithelium.

# Materials and methods

## Key resources table

| Reagent type (species) or resource | Designation | Source or reference | Identifiers | Additional information |
|---|---|---|---|---|
| Strain, strain background (*Xenopus laevis* (*Female*), *Oocyte positive*, Pigmented) | *Xenopus laevis* | Nasco | Cat #: LM00531 | |
| Strain, strain background (*Xenopus laevis* (*Female*), *Oocyte positive*, Albino) | *Xenopus laevis* | Nasco | Cat #: LM00531(A) | |
| Strain, strain background (*Xenopus laevis* (*Male*), Mature 7.5–9 cm, Pigmented) | *Xenopus laevis* | Nasco | Cat #: LM00715 | |
| Genetic reagent (Anillin morpholino oligonucleotide) | Anillin MO | Gene Tools; (*Reyes et al., 2014*); https://doi.org/10.1016/j.cub.2014.04.021 | | An antisense MO (Gene Tools) was generated to target the 5′UTR of *Xenopus laevis* anillin with the sequence 5' – TGGCTAGTAACTCGATCCTCAGACT – 3'. |
| Antibody (anti-Anillin antibody) | α-Anillin | gift from Aaron Straight, Stanford University (*Straight et al., 2005*) https://doi.org/10.1091/mbc.e04-08-0758 | | 1:500 dilution in Tris-buffered saline (50 mM Tris and 150 mM NaCl [pH 7.4]) containing 10% fetal bovine serum (10082–139; Invitrogen), 5% DMSO and 0.1% NP-40 overnight at 4°C |
| Antibody (goat anti-rabbit-Alexa Fluor 488) | | Life Technologies | Cat #: A11008; Lot: 1583138 | 1:200 dilution in Tris-buffered saline (50 mM Tris and 150 mM NaCl [pH 7.4]) containing 10% fetal bovine serum (10082–139; Invitrogen), 5% DMSO and 0.1% NP-40 overnight at 4C |
| Recombinant DNA reagent (pCS2+/Anillin) | Anillin | (*Reyes et al., 2014*); https://doi.org/10.1016/j.cub.2014.04.021 | | |
| Recombinant DNA reagent (pCS2+/Anillin-3xGFP) | Anillin-3xGFP | (*Reyes et al., 2014*); https://doi.org/10.1016/j.cub.2014.04.021 | | |

*Continued on next page*

*Continued*

| Reagent type (species) or resource | Designation | Source or reference | Identifiers | Additional information |
|---|---|---|---|---|
| Recombinant DNA reagent (pCS2+/Anillin-3xmCherry) | Anillin-3xmChe | (*Reyes et al., 2014*); https://doi.org/10.1016/j.cub.2014.04.021 | | |
| Recombinant DNA reagent (pCS2+/Actin-mNeon) | Actin-mNeon | this paper | | *Xenopus laevis* actin was cloned from a cDNA library generated from stage 35 tadpoles (*Higashi et al., 2016*) https://doi.org/10.1016/j.cub.2016.05.036.actin into pCS2+ with the following primers: S: aaaaGAATTCaatgga agacgatattgccgcactg AS: ttttTCTAGAttagaagcatt tacggtggacaattgagg |
| Recombinant DNA reagent (pCS2+/Shroom3) | Shroom3 | gift from Sergei Sokol, Icahn School of Medicine at Mount Sinai (*Chu et al., 2013*); https://doi.org/10.1371/journal.pone.0081854 | | |
| Recombinant DNA reagent (pCS2+/Lifeact-RFP) | Lifeact-RFP | (*Higashi et al., 2016*); https://doi.org/10.1016/j.cub.2016.05.036 | | |
| Recombinant DNA reagent (pCS2+/Lifeact-GFP) | Lifeact-GFP | (*Higashi et al., 2016*); https://doi.org/10.1016/j.cub.2016.05.036 | | |
| Recombinant DNA reagent (pCS2+/BFP-membrane) | BFP-membrane | (*Higashi et al., 2016*); https://doi.org/10.1016/j.cub.2016.05.036 | | |
| Recombinant DNA reagent (pCS2+/E-cadherin-3xGFP) | E-cadherin-3xGFP; E-cad-3xGFP | (*Higashi et al., 2016*); https://doi.org/10.1016/j.cub.2016.05.036 | | |
| Recombinant DNA reagent (pCS2+/mCherry-α-catenin) | mChe-α-catenin | (*Higashi et al., 2016*); https://doi.org/10.1016/j.cub.2016.05.036 | | |
| Recombinant DNA reagent (TOPO-SF9-YFP) | | gift from E.M. Munro, University of Chicago (*Hashimoto et al., 2015*); https://doi.org/10.1016/j.devcel.2014.12.017 | | |
| Recombinant DNA reagent (pCS2+/SF9-mNeon) | SF9-mNeon | this paper | | SF9 was subcloned from TOPO into pCS2 + with the following primers: S: AAAAGGATCCACCA TGGCCGAGGTGCAGC AS: TTTTATCGATTACCTA GGACGGTCAGCTTGG |

*Continued on next page*

*Continued*

| Reagent type (species) or resource | Designation | Source or reference | Identifiers | Additional information |
|---|---|---|---|---|
| Recombinant DNA reagent (pCS2+/Vinculin-mNeon) | Vinculin-mNeon | this paper | | *Xenopus laevis* vinculin was subcloned using BamHI and XbaI restriction enzymes from pCS2+/Vinculin-3xGFP from (*Higashi et al., 2016*); https://doi.org/10.1016/j.cub.2016.05.036 |
| Commercial assay or kit (mMESSAGE mMACHINE SP6 Transcription Kit) | mMESSAGE mMACHINE SP6 Transcription Kit | Thermo Fisher Scientific | AM1340 | |
| Chemical compound, drug (Jasplakinolide) | Jasplakinolide; Jas | Cayman Chemical | Cat #: 11705, CAS: 102396-24-7 | |
| Chemical compound, drug (phalloidin Alexa Fluor 568) | phalloidin | Life Technologies | Cat #: A12380, Lot: 1154065 | 1:100 dilution in Tris-buffered saline (50 mM Tris and 150 mM NaCl [pH 7.4]) containing 10% fetal bovine serum (10082–139; Invitrogen), 5% DMSO and 0.1% NP-40 overnight at 4C |
| Chemical compound, drug (BODIPY FL phallacidin) | BODIPY FL phallacidin | Thermo Fisher Scientific | Cat #: B607 | |
| Chemical compound, drug (ATP) | ATP | Sigma | Cat #: A2383-5G; Lot: SLBD2725V | |
| Software, algorithm (Graphpad Prism 6.01) | | GraphPad Software, La Jolla California USA, www.graphpad.com | | |
| Software, algorithm (Fiji (ImageJ)) | | (*Schindelin et al., 2012*) https://doi.org/10.1038/nmeth.2019 | | |
| Software, algorithm (Custom plugin for ImageJ that uses bUnwarp J plugin) | Strain mapping | (*Feroze et al., 2015*) https://doi.org/10.1016/j.ydbio.2014.11.011 | | |

## *Xenopus laevis* embryos and microinjections

All studies conducted using *Xenopus laevis* embryos strictly adhered to the compliance standards of the US Department of Health and Human Services Guide for the Care and Use of Laboratory Animals and were approved by the Institutional Animal Care and Use Committees at the University of Michigan or the University of Pittsburgh. *Xenopus* embryos were collected, in vitro fertilized, de-jellied, and microinjected with mRNAs for fluorescent probes using methods described previously (*Reyes et al., 2014*). Embryos were injected at either the two-cell (each cell injected twice) or the four-cell stage (each cell injected once) and allowed to develop to gastrula stage (Nieuwkoop and Faber (NF) stages 10 to 11) or neurula stage (NF stage 15). For gastrula-stage experiments, embryos were injected at the animal hemisphere at the two-cell stage, injecting each cell twice, or the four-cell stage, injecting each cell once. For neurula-stage experiments, embryos were injected at the equator at the four-cell stage, injecting two of the cells to characterize F-actin defects and all four of the cells for tissue stiffness measurements. Amounts of probes per single injection are as follows:

Actin-mNeon, 17 pg or 36 pg when anillin was knocked down; Anillin untagged overexpression (OE), 70 pg; Anillin-3xmCherry, 70 pg; Anillin-3xGFP or 3xmCherry OE, 300 pg; mCherry-α-catenin, 30 pg; E-cadherin-3xGFP, 50 pg; Lifeact-GFP or RFP, 16 pg; membrane-TagBFP, 60 pg; SF9-mNeon, 74 pg; Shroom3 OE, 200 pg; Anillin morpholino 211 pg. Anillin MO was injected similar to (*Reyes et al., 2014*) except instead of injecting 5 nl of 5 mM morpholino, here we injected 10 nl of 2.5 mM morpholino to make injections easier, as diluting the morpholino made it less viscous and easier to inject.

## Constructs

*Xenopus laevis* actin was cloned from a cDNA library generated from stage 35 tadpoles (*Higashi et al., 2016*) and verified by sequencing. pCS2+/Shroom3 was a generous gift from Sergei Sokol. SF9 was a generous gift from Ed Munro (*Hashimoto et al., 2015*) and was subcloned into pCS2+/mNeon. All other constructs were in pCS2+ plasmid, and mRNA was synthesized as previously reported (*Reyes et al., 2014*; *Higashi et al., 2016*).

## Experimental replicates

Multiple embryos were used for all experimental replicates. Each experiment was conducted in three separate replicates (embryos isolated from three separate frogs) except the following, which were conducted with two replicates: Laser ablation with jasplakinolide treatment, laser ablation with anillin knockdown and jasplakinolide treatment, observing anillin localization in embryos with Shroom3 overexpression, dorsal isolate characterization when anillin was knocked down, dorsal isolate tissue stiffness measurements when anillin was knocked down.

## Confocal microscopy

Images of gastrula-stage embryos were collected on an inverted Olympus Fluoview 1000 microscope equipped with a 60X super corrected PLAPON 60XOSC objective (numerical aperture [NA]=1.4; working distance = 0.12 mm) and FV10-ASW software as previously described in (*Reyes et al., 2014*).

Images of dorsal isolates were collected using a confocal laser scan head (SP5 Leica Microsystems) mounted on an inverted compound microscope (DMI6000, Leica Microsystems) equipped with a 25X water immersion objective using acquisition software (LASAF, Leica Microsystems).

## Laser ablation

Single plane movies were collected on a Leica Inverted SP5 Confocal Microscope System with 2-Photon using a 60X objective. White-light laser was tuned for emission at 488 nm and or 561 nm to visualize GFP or mCherry, respectively. Junction cuts were made with the 2-Photon laser with a region of interest (ROI) of 0.36 × 4 μm. Medial-apical cuts were made on embryos expressing E-cadherin-3xGFP and Lifeact-RFP to mark the medial-apical surface. Medial-apical cuts were made using the 2-Photon laser with a ROI of 1 × 15 μm. Each embryo was only cut three times, and cuts were made in distant areas of the embryo.

## Fixed staining and immunofluorescence

Gastrula-stage embryos were immunostained using methods described previously (*Reyes et al., 2014*) with the following changes: embryos were fixed with 1.5% formaldehyde, 0.25% glutaraldehyde, 0.2% Triton X-100, and 0.88X MT fix buffer (1X MT buffer: 80 mM K-PIPES, 5 mM EGTA, 1 mM MgCl$_2$ [pH 6.8]) and blocked in Tris-buffered saline (50 mM Tris and 150 mM NaCl [pH 7.4]) containing 10% fetal bovine serum (10082–139; Invitrogen), 5% DMSO. and 0.1% NP-40 overnight at room temperature. Embryos were stained with 1:100 and phalloidin Alexa Fluor 568 (Life Technologies A12380 Lot: 1154065), and 1:500 anti-Anillin antibody (generously provided by Aaron Straight, Stanford University (*Straight et al., 2005*)), which was detected by 1:200 goat anti-rabbit-Alexa Fluor 488 (Life technologies A11008 Lot: 1583138).

## ATP experiments

Whole gastrula-stage embryos (NF stage 10.5) were imaged on an Olympus SZX16 Fluorescent Stereoscope, and 320 nl of 50 mM ATP (Sigma A2383-5G Lot: SLBD2725V) diluted in 0.1X MMR was

added to one side of the embryo with a microinjection needle. For cell view experiments, gastrula-stage embryos were mounted in 0.1X MMR for confocal imaging in an imaging slide with an opening in the top of the chamber. During live imaging, 100 µl of 500 µM ATP diluted in 0.1X MMR was added to the chamber.

## Jasplakinolide experiments

For confocal time-lapse imaging, gastrula-stage embryos were mounted in 0.1X MMR for confocal imaging in an imaging slide with an opening in the top of the chamber. During live imaging, 100 µl of 20 µM Jasplakinolide diluted in 0.1X MMR was added to the chamber. For laser ablation experiments, embryos were added to a microcentrifuge tube with 20 µM Jasplakinolide diluted in 0.1X MMR. After 1 hr of incubation, embryos were mounted on a slide and imaged.

## F-actin localization in dorsal isolates

Anillin morpholino was co-injected with Alexa Fluor 647-conjugated dextran into one dorsal blastomere at the four-cell stage. At NF stage 15, dorsal isolates were microsurgically isolated, fixed in 4% paraformaldehyde for 4 hr, bisected, and stained for F-actin (BODIPY FL phallacidin, Thermo Fisher Scientific B607). Some explants were not bisected, but instead the neural ectoderm was imaged *en face*.

## Dorsal isolate stiffness

Anillin morpholino or water (control) was injected into each dorsal blastomere at the four-cell stage. Dorsal isolates were isolated from embryos at NF stage 15 and loaded into the nanoNewton Force Measurement Device to measure time-varying elastic modulus using a stress-relaxation protocol (*Davidson and Keller, 2007*). In brief, the tissue sample is compressed along its anterior-posterior axis, while the resistive force is measured by the deflection of an optical fiber force transducer. Young's modulus after 180 s of compression is then calculated using measured resistive force, explant cross-sectional area, and strain measurements.

## Image analysis

Vinculin and α-catenin intensities: A bicellular junction was traced from vertex to vertex with a segmented line five pixels wide in Fiji (*Schindelin et al., 2012*). Vinculin and α-catenin intensities were normalized by dividing by the membrane probe (mem-BFP) intensity. For perpendicular vs. parallel actin fiber experiments, Lifeact-RFP was expressed to visualize actin. Two junctions were chosen for each image by looking at the Lifeact-RFP channel for bundles of F-actin associated with junctions in the two orientations. Vinculin and membrane intensities were then measured as above.

Vertex recoil after laser ablation: In Fiji, a straight line was drawn between the vertices to measure the amount of separation over time.

Percentage of cells that separate perpendicularly: Cells that displayed junctional splaying of E-cadherin-3xGFP were marked as exhibiting perpendicular separation. Each data set was qualitatively examined three times for this phenotype, and the percentages were averaged.

F-actin and myosin II intensity: In Fiji, a circular ROI with a diameter of 10 µm placed in the center of the cell was used to measure medial-apical intensity. For fixed phalloidin staining of F-actin, the raw intensity values were used. For live imaging of myosin II with SF9-mNeon, the intensity was normalized by dividing by the membrane probe intensity.

Medial-apical recoil after laser ablation: In Fiji, a straight line was drawn spanning the junction-junction distance perpendicular or parallel to the cut site to measure the amount of apical expansion over time.

Strain mapping: Deformations of cells were detected by adapting an algorithm, which registers two images by deforming one image to match the other (*Sorzano et al., 2005*). Single-plane confocal images at two time points were aligned (*Thévenaz et al., 1998*) and analyzed with the bUnwarpJ plug-in for ImageJ. The deformation between the two images is reported as a directionless pixel intensity based on the distance each pixel had to be manipulated in x and y to be realigned to fit the image. For image alignment, the ablation site was masked in Image J, as these areas can have wounding artifacts that are static and impact the strain output. Ablation sites were masked by creating a new slice for the image, selecting the ablated area, setting it the value to 0 (Math, Set), and

setting the outer area to 255. Tiff images with masked ablation sites were then processed with bUnwarpJ. Strains ($\varepsilon_x$, $\varepsilon_y$, $\varepsilon_{yx}$, $\varepsilon_{yx}$) were calculated first in the image directions x and y. Principle strains ($\varepsilon_1$ and $\varepsilon_2$) were calculated to minimize shear strain, and their contribution to strain area ($\varepsilon_1 + \varepsilon_2 + \varepsilon_1 \varepsilon_2$) was calculated and displayed as a strain map. Strain maps in *Figure 3F* are images with a Phase LUT applied in ImageJ.

ATP contraction index: Kymographs were generated in Fiji, and shift of the pigment in each kymograph was measured three times as length by tracing a line across the image: near the top, center, and bottom of the kymograph. These values were averaged, and movement of the whole embryo in the field of view was subtracted from the contraction by tracing the bottom edge of the embryo in the kymograph to obtain the contraction index.

F-actin intensity after ATP: In Fiji, a circular ROI with a diameter of 10 µm placed in the center of the cell was used to measure medial-apical intensity over time. Change in F-actin intensity was measured by subtracting baseline intensity from peak intensity. Baseline was the average of the 10 time points before ATP, and peak was the average of the 10 time points starting at 45 s after ATP. For near junction vs. center of cell F-actin intensity comparisons, a smaller ROI with a diameter of 3 µm was used to measure F-actin intensity in the center of the cell or near the cell periphery.

Blinded categorization of medial-apical F-actin organization: Confocal images of the F-actin channel (Lifeact-RFP) when Anillin FL or anillin mutants were overexpressed were exported as JPEGs using ImageJ. File names were then randomly generated using the Random Names program created by Jason Faulkner from How-To-Geek https://goo.gl/1EcGCa. Cells that were ~80% or more visible in the field of view were then categorized as: (1) no F-actin fibers, (2) short F-actin fibers that do not span the apical surface, or (3) long F-actin fibers that span the surface. Each cell could only be placed into one category, so if for example a cell contained both short and long fibers it would go into the long fiber category. Categorization was done by three different blinded individuals, and their categorizations were averaged to produce the final graph.

Anillin medial-apical intensity: A circular ROI with a diameter of 10 µm was placed in the center of the cell, and average intensity was collected for Anillin FL or anillin mutant constructs tagged with 3xGFP. These values were not normalized, but the same excitation laser intensity was used for acquisition across experiments.

FRAP: FRAP data were collected and analyzed as described in (*Higashi et al., 2016*) with the following modifications: only a single plane was captured instead of Z-stacks. Medial-apical Actin-mNeon was bleached instead of junction proteins. Curves were fit with a double association curve in Prism6.

## Statistical analysis

Unpaired t-tests were performed using GraphPad Prism version 6.01 for Windows, GraphPad Software, La Jolla California USA, www.graphpad.com. Double exponential curves with the following constraints, Y = 0 and plateau must be <1, were fit to FRAP data in GraphPad Prism. 2-way ANOVA was performed using IBM SPSS Statistics for Windows, Armonk, NY: IBM Corp.

## Acknowledgements

We thank EM Munro for SF9 myosin II intrabody construct, S Sokol for the Shroom3 construct, and AF Straight for the anti-Anillin antibody. Members of the ALM and LAD lab for reviewing the manuscript and providing useful discussion and critical feedback. AC Martin for helpful discussion and critical feedback on experiments. Saranyaraajan Varadarajan, Shahana Chumki, and Lauren Smith for blinded characterization of data. Biomedical Research Core Facilities at UM for technical support and access to the Leica Inverted SP5 Confocal Microscope System with 2-Photon.

## Additional information

### Funding

| Funder | Grant reference number | Author |
| --- | --- | --- |
| National Science Foundation | Graduate Research Fellowship | Torey R Arnold Rachel E Stephenson |

| | | |
|---|---|---|
| Rackham Graduate School | Rackham Merit Fellowship | Torey R Arnold |
| National Institutes of Health | Biomechanics in Regenerative Medicine Training Grant | Joseph H Shawky |
| Arnold and Mabel Beckman Foundation | Beckman Scholars Fellowship | Kayla M Dinshaw |
| Japan Society for the Promotion of Science | Postdoctoral Fellowship | Tomohito Higashi |
| Eunice Kennedy Shriver National Institute of Child Health and Human Development | R01 HD04475 | Lance A Davidson |
| National Heart, Lung, and Blood Institute | R56 HL134195 | Lance A Davidson |
| National Institute of General Medical Sciences | R01 GM112794 | Ann L Miller |

The funders had no role in study design, data collection and interpretation, or the decision to submit the work for publication.

### Author contributions

Torey R Arnold, Conceptualization, Investigation, Methodology, Writing—original draft, Writing—review and editing, Plasmid cloning, Additional funding resources; Joseph H Shawky, Rachel E Stephenson, Investigation, Writing—review and editing, Additional funding resources; Kayla M Dinshaw, Plasmid cloning, Additional funding resources; Tomohito Higashi, Methodology, Writing—review and editing, Additional funding resources; Farah Huq, Plasmid cloning; Lance A Davidson, Software, Methodology, Writing—review and editing, Additional funding resources; Ann L Miller, Conceptualization, Supervision, Funding acquisition, Methodology, Writing—review and editing

### Author ORCIDs

Torey R Arnold http://orcid.org/0000-0002-1663-1953
Tomohito Higashi https://orcid.org/0000-0001-5616-1477
Lance A Davidson http://orcid.org/0000-0002-2956-0437
Ann L Miller http://orcid.org/0000-0002-7293-764X

### Ethics

Animal experimentation: All studies conducted using Xenopus laevis embryos strictly adhered to the compliance standards of the US Department of Health and Human Services Guide for the Care and Use of Laboratory Animals and were approved by the University of Michigan's Institutional Animal Care and Use Committee (PRO00007339) or the University of Pittsburgh's Institutional Animal Care and Use Committee (Protocol #18022377).

### Decision letter and Author response

Decision letter https://doi.org/10.7554/eLife.39065.034
Author response https://doi.org/10.7554/eLife.39065.035

## Additional files

### Supplementary files

• Transparent reporting form
DOI: https://doi.org/10.7554/eLife.39065.032

### Data availability

All data generated or analysed during this study are included in the manuscript and supporting files. Source data files have been provided for: Figures 1, 2, 3, 4, 6, 7 and 8.

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
