## [Decision Letter]

Thank you for submitting your article "Anillin regulates epithelial cell mechanics by structuring the medial-apical actomyosin network" for consideration by *eLife*. Your article has been reviewed by three peer reviewers, one of whom is a member of our Board of Reviewing Editors, and the evaluation has been overseen by Anna Akhmanova as the Senior Editor. The following individual involved in review of your submission has agreed to reveal his identity: Alpha Yap (Reviewer #3).

The reviewers have discussed the reviews with one another and the Reviewing Editor has drafted this decision to help you prepare a revised submission.

Summary:

This interesting manuscript addresses the issue of how acto-myosin regulation influences tissue mechanics. Using the early *Xenopus* embryo, the authors show that anillin supports a medial-apical actomyosin network in addition to actomyosin at junctions, which they reported earlier (Reyes et al., 2014). The reviewers agreed that your manuscript is polished, complete and clear. They also agreed that most of the conclusions presented by the authors are well-substantiated.

The major concerns raised by the reviewers relates to the axis of force generated by anillin. The hypothesis of two different axes is possible, but is not really tested. This is for two reasons: a) It assumes that vinculin is responding to a perpendicular force. While this is how we draw the cartoons, that is purely speculative. Most forces will have some perpendicular and shear (parallel) components. b) It depends on interpreting differences in recoil as differences in tension. Recoil will also have a frictional element and thus the apparent contradiction needs to have experiments coupled possibly with mechanical modelling.

Essential revisions:

The reviewers recommended additional experiments and textual changes for your manuscript:

A) Additional analyses and experiments

1) As the authors cautiously point out in the first paragraph of the subsection “Anillin increases junctional tension but reduces recoil of junction vertices after laser ablation”, depleting anillin not only reduces vinculin tension-sensor signal but also α-catenin, confounding the conclusions about tension-sensor levels. The authors should present a ratio of tension-sensor signal to α-catenin or total vinculin signal to normalize for junctional integrity/robustness/mass. This will allow them to conclude more confidently whether there is a reduction in tension in this condition, since the lower tension-sensor signal could be due to lower tension-sensor occupancy of junctions following anillin depletion.

The shape of the recoil curves in Figures 1D and 6F are unusual, exhibiting a delay after cutting. Do individual cut recoils exhibit a rapid increase and gradual plateau? If not, perhaps the unusual shape of these recoil dynamics is a signature of the complex directionality of forces on the junction. A specific question about the unusual recoil kinetics is what happens between 20 and 46 seconds for a junction like the one shown in Figure 6G, which has totally opened (parallel) at 46 seconds, while the population average at 20 seconds in 6F shows only 0.2 microns of parallel recoil?

2) The phenomenon reported in Figures 1F and 6G of perpendicular junctional dissociation needs to be quantified. Is there any separation in this perpendicular direction in control junctions or those depleted of anillin? These measurements for the population of cut junctions need to be reported. In addition, a control example, with comparable time-course, needs to be shown to accompany abnormal junction response.

3) The contribution of anillin's various protein interaction domains is implicated by expressing truncation constructs and assessing apical actin network organization. However, as the authors report, many of the anillin truncations are not properly enriched in the apical actin network. Just as low expression level for a truncation construct prohibits the attribution of any failure of that truncation to replace function of full-length, the same is true if the truncation is not present/enriched in the subcellular structure of interest. The authors need to tone down their conclusions of the roles of anillin's various regions in organizing apical actin for constructs that do not normally localize. I would like to see both apical *enrichment* (numbers in 5C and 5G compared to cytoplasmic or other intensity?) and network organization quantified (likely with an open-source image analysis software) so that normalizing for anillin abundance could be attempted/done.

4) Does anillin contribute differentially to tension/contractility and stiffness in the medial-apical network? Here it might be informative to manipulate the medial-apical region directly, by making laser cuts in the medial apical cortex itself (e.g. Stephan Grill and the 1-cell *C. elegans* embryo).

5) The reviewers wondered: Is the medial-apical network pulsatile? Is pulsatility affected by anillin overexpression (is it dampened)? If possible, measuring any pulses in the network contraction would allow the authors to add this as a part of their model for how anillin works.

6) Analysis of actin stability in the overexpression of the delta-F-actin mutant should be performed.

B) Textual changes:

1) Introduction. The last two paragraphs are a little long and discursive. In particular, the final paragraph rather anticipates the story in too much detail.

2) Figure 7: morphological defects seen in presomitic mesoderm, notochord and ectoderm in KD – please describe these.

C) If possible, the reviewers thought that a mechanical model could add to the interpretation of the data. They agreed that some of the experiments above may address these concerns as well. I think that it is very difficult to interpret these results without a mechanical model (or at least I find it very difficult). There are at least 3 levels of complexity that need to be considered. 1) Actomyosin can contribute to tension but also influence viscoelasticity. (Conceivably, a highly cross-linked actomyosin network might be very elastic but not actually generate much tension.) 2) As the authors note (and has been developed in some of e.g. Pierre-Francois Lenne's reviews) recoil is the product of tension divided by friction. It can't always be equated to tension; and this is potentially one of those cases, since the authors data indicate that tissue stiffness is increased. 3) Both the medial-apical and junctional networks may contribute and it is difficult to tease these apart. Looking at their videos, I would consider the possibility that the impact of anillin on the medial-apical network may be principally to increase stiffness in this region of the cell, rather than as a source of tensile force. This could explain the slower recoil of junctions when anillin is overexpressed, because a stiffer medial-apical network limits recoil of the junction. Of course, this implies that the junctions and medial-apical network are physically connected.

Also the reviewers agreed that this would help the manuscript, the authors thought that if this was not able to be added in the two month time frame, the experiments requested may also address these same concerns.

---

## [Author Response]

The major concerns raised by the reviewers relates to the axis of force generated by anillin. The hypothesis of two different axes is possible, but is not really tested. This is for two reasons: a) It assumes that vinculin is responding to a perpendicular force. While this is how we draw the cartoons, that is purely speculative. Most forces will have some perpendicular and shear (parallel) components. b) It depends on interpreting differences in recoil as differences in tension. Recoil will also have a frictional element and thus the apparent contradiction needs to have experiments coupled possibly with mechanical modelling.

We would like to thank the reviewers for their concerns related to the axes of force generation. We agree with the reviewers that the original submission did not fully test whether the effects Anillin has on medial-apical actomyosin increase perpendicular forces exerted on cell-cell junctions. To address this concern, we performed two new experiments.

First, we examined the orientation and connection of the medial-apical F-actin bundles to junctions for insights into the contractile nature of the actin fibers. We examined junctional Vinculin intensity in cells overexpressing Anillin and compared Vinculin recruitment at junctions with F-actin bundles oriented *perpendicular* or *parallel* to the junction. We found that junctions with *perpendicularly* oriented medial-apical F-actin fibers exhibited higher Vinculin intensity compared to junctions with *parallel* F-actin bundles. This data is now presented in Figure 3A and B.

Second, we probed the mechanics of the apical surface of cells by making laser cuts in the medial-apical cortex. Epithelial cells store tensile energy through the strain of the elastic material of the cortex. By measuring apical expansion of epithelial cells after medial-apical laser ablation, we found that the tensile energy stored in the medial-apical cortex (indicated by strain released after ablation) increased when Anillin was overexpressed and decreased when Anillin was knocked down. Strain intensity maps (determined by fitting the final ablation time course image to the first image) indicate the strain intensity for each pixel as well as areas of expansion and contraction following laser ablation. Local cellular strain released by ablation was reduced when Anillin was knocked down and increased when Anillin was overexpressed, while tissue-level strains were similar for controls and Anillin overexpression and reduced for Anillin knockdown. This new data presented in Figure 3C-G.

Together, the Vinculin recruitment and medial-apical laser ablation results indicate that tensile energy is stored across the apical surface of *Xenopus* epithelial cells, and Anillin organizes medial-apical actomyosin into a load-bearing structure. While we do not rule out the possible contributions that stiffness or friction may play in the mechanical properties of this epithelial tissue, we think this combination of experiments supports the conclusion that Anillin modulates the tensile energy stored (and recoil transmitted) across the apical surface of epithelial cells after laser ablation.

Essential revisions:The reviewers recommended additional experiments and textual changes for your manuscript:A) Additional analyses and experiments1) As the authors cautiously point out in the first paragraph of the subsection “Anillin increases junctional tension but reduces recoil of junction vertices after laser ablation”, depleting anillin not only reduces vinculin tension-sensor signal but also α-catenin, confounding the conclusions about tension-sensor levels. The authors should present a ratio of tension-sensor signal to α-catenin or total vinculin signal to normalize for junctional integrity/robustness/mass. This will allow them to conclude more confidently whether there is a reduction in tension in this condition, since the lower tension-sensor signal could be due to lower tension-sensor occupancy of junctions following anillin depletion.

We thank the reviewers for suggesting that we include a ratio of Vinculin to α-catenin to help clarify whether there is a reduction in tension when Anillin is knocked down. Ratio and intensity scatter plots are shown below. Previously published data from our lab (Higashi et al., 2016) and Figure 1—figure supplement 1A-C are also provided for comparison. The Vinculin/α-catenin ratio (see first graph on left) shows a similar range but statistically significant increase for Anillin overexpression compared to controls. However, for Anillin knockdown, the Vinculin/α-catenin ratio is ~7x greater than controls. We believe this is because the normalized (compared to membrane) α-catenin intensity is near zero for many of the intensity measurements, while Vinculin intensity clusters around 0.5 (see graph on right). This results in a very high Vinculin/α-catenin ratio which could be interpreted as 1) dramatically increased tension or 2) reduced tension, where although there is less overall tension being applied to the junction, each remaining α-catenin is under increased load (as there are fewer α-catenins to bear the force), and thus the remaining α-catenins may be maximally saturated with Vinculin. Based on our previous results showing that when tension is increased by the cytokinetic contractile ring, the Vinculin/α-catenin ratio increases about 1.2x as the cleavage furrow ingresses (Higashi et al., 2016), we do not think the abnormally high Vinculin/α-catenin ratio (7x increase) when Anillin is knocked down represents increased tension. Therefore, we decided to present the Vinculin and α-catenin data in two separate graphs in Figure 1—figure supplement 1B, C so that the reader can appreciate that α-catenin is reduced in Anillin knockdown but unchanged in Anillin overexpression compared with controls. We have also modified the text to make it clear that we can only conclude that these data suggest increased tension when Anillin is overexpressed, but the data are inconclusive about the amount of tension when Anillin is knocked down. If the reviewers think it is necessary, we could include the Vinculin/α-catenin ratio graph or scatter plots as part of Figure 1—figure supplement 1.

**Author response image 1. respfig1:** Ratio of junctional Vinculin to α-catenin when Anillin is perturbed. (Left) Ratio of Vinculin to α-catenin. Measurements were taken by tracing a bicellular junction from vertex to vertex and normalizing Vinculin and α-catenin intensity to membrane intensity. Error bars, S.E.M, n=number of junctions, controls=45, Anillin KD=31, Anillin OE=54. Statistics, unpaired t-test. (Right) Scatter plotof Vinculin and α-catenin intensity, each normalized to membrane signal.

The shape of the recoil curves in Figures 1D and 6F are unusual, exhibiting a delay after cutting. Do individual cut recoils exhibit a rapid increase and gradual plateau? If not, perhaps the unusual shape of these recoil dynamics is a signature of the complex directionality of forces on the junction.

We agree with the reviewers that our recoil graphs look different from other published junction recoil dynamics. One reason for this is, as the reviewers noted, we chose to focus on the early portion of recoil because we found, at least in our system, that this is where the reproducible data about relative tension of the junctions is found. Shortly after 18 s, many of the cells begin to expel their cytoplasmic contents, and shortly after that, the surrounding cells can begin to extrude the ablated cells. Therefore, we chose to focus on the early recoil events that occur just after ablation. Five randomly selected individual recoil examples from each data set are shown in Figure 1—figure supplement 1D. You can appreciate that there are several examples of delayed recoil (one in control, one in knockdown, and in all of the overexpressing examples), while the rest of the junctions recoil instantly. We are not sure what the cause of this delay is, but since the delay is commonly seen in Anillin overexpression, it may result from the complex directionality of the forces acting on the junctions, as the reviewers suggest, or from other effects that Anillin may have on cellular mechanics such as junctional stiffness.

Additionally, we were able to track recoil of vertices for the dataset out to 45 s and have included this graph below and in Figure 1—figure supplement 1E. This data shows the same trend where Anillin knockdown cells recoil more compared to controls, and Anillin overexpression cells recoil less. The curves appear to begin plateauing near ~40 s. However, accurate tracking of the junction vertices past 45 s was not possible because expelled cytoplasmic contents obstructed the field of view, or surrounding cells began to extrude the ablated cells.

A specific question about the unusual recoil kinetics is what happens between 20 and 46 seconds for a junction like the one shown in Figure 6G, which has totally opened (parallel) at 46 seconds, while the population average at 20 seconds in 6F shows only 0.2 microns of parallel recoil?

We thank the reviewer for this question. We would like to clarify that junction recoil was not measured from the “tips” of the disintegrating junction, as this would be measuring both recoil and the disassembly of the junction. Instead, junction recoil was measured as the change in the position of the vertices after ablation. Thus, at 20 s, the junction in Previous Figure 6 noted by the reviewer has only recoiled 0.14 µm. See annotated figure (Author response image 2 -previous Figure 6G from initial submission). We have now clarified how measurements were made throughout the figures by using red lines to indicate the initial position of the junction vertices (see updated Figure 7G), and when associated with a graph measuring vertex recoil, we placed green dots directly on the vertices and indicated in the figure legends that these were the sites measured (see updated Figure 1C).

**Author response image 2. respfig2:** Previous Figure 6G: Confocal images of an embryo expressing E-cad-3xGFP and treated with 20 µm jasplakinolide before and after laser ablation. Blue boxes show the zoomed area for the ablation montage. Green dashed lines indicate the position of the vertices relative to the edge of the image, orange dashed line indicates the perpendicular separation between the two cells, grey dashed line indicates the space forming between the two cells, and blue arrows represent forces on junctions adjacent to the lower cell vertex. Notice that the lower vertex only begins to separate in the parallel direction after the forces perpendicular to the adjacent junction (blue arrows) lead to loss of adhesion between the two cells.

2) The phenomenon reported in Figures 1F and 6G of perpendicular junctional dissociation needs to be quantified. Is there any separation in this perpendicular direction in control junctions or those depleted of anillin? These measurements for the population of cut junctions need to be reported. In addition, a control example, with comparable time-course, needs to be shown to accompany abnormal junction response.

We thank the reviewers for the suggestion of quantifying the number of cells that recoil perpendicularly when their cell-cell junctions are ablated. We agree that it would be informative to quantify the distance of perpendicular separation in cells that exhibit this type of recoil; however, because E-cadheren-3xGFP was the only fluorescent protein expressed (in addition to Anillin-3xmCherry) in these experiments, we were not able accurately track this separation over time. This is because the junctions disintegrate over time after junctional laser ablation, making measurements of perpendicular separation in controls and Anillin knockdown (where no other fluorescent probe were expressed) inaccurate.

However, we were able to re-analyze the movies and count the instances of E-cadherin “splaying”, which we used as a readout for occurrence of perpendicular separation. E-cadherin splaying is defined as when the stubs of the junction left after laser ablation were separated in two. We found that control and Anillin overexpressing cells exhibited E-cadherin splaying about 55-60% of the time, whereas Anillin knockdown cells only showed splaying about 10% of the time. This data is now included in Figure 1—figure supplement 2B, C. Additionally, we added control time-course examples to accompany perpendicular separation data in Figure 1—figure supplement 2 and Figure 7—figure supplement 1.

3) The contribution of anillin's various protein interaction domains is implicated by expressing truncation constructs and assessing apical actin network organization. However, as the authors report, many of the anillin truncations are not properly enriched in the apical actin network. Just as low expression level for a truncation construct prohibits the attribution of any failure of that truncation to replace function of full-length, the same is true if the truncation is not present/enriched in the subcellular structure of interest. The authors need to tone down their conclusions of the roles of anillin's various regions in organizing apical actin for constructs that do not normally localize. I would like to see both apical enrichment (numbers in 5C and 5G compared to cytoplasmic or other intensity?) and network organization quantified (likely with an open-source image analysis software) so that normalizing for anillin abundance could be attempted/done.

We thank the reviewers for pointing out that both local enrichment as well as protein interactions could impact Anillin’s ability to organize medial-apical F-actin into long bundles. Although it is a good suggestion, unfortunately we could not normalize the medial-apical Anillin signal to cytoplasmic intensity, as only the apical 2.5 µm of cells was imaged. Additionally, Anillin also localizes to the nucleus, so a full z-scan of the cells (~20 µm) would be required. Imaging this deeply is not possible with our live imaging setup, as the opaque nature of frog embryo cells mostly blocks light transmission at ~10 µm. In lieu of this, we have toned down our conclusions about the roles of Anillin’s binding regions in organizing apical actin in the Results section, particularly for the ΔC2 and ΔPH mutants, which did not localize with the same intensity as Anillin FL. The Δact mutant localized with the same intensity as the Δmyo mutant, which was able to form long bundles, so we argue that the inability of ∆act to promote long F-actin bundles suggests that Anillin’s actin-binding function is important.

To characterize the F-actin bundles, we attempted to use several open-source analysis plugins for ImageJ including “Skeleton” and “Ridge finder” but had little success. The noise to actin bundle ratio was often quite high, and these software options could not detect bundles that were easily detected by eye. Additionally, in control cells, medial-apical bundles were sometimes detected when they weren’t present. This led us to use the blinded fiber classification approach, where the data was blinded for the user, and the organization of the medial-apical actin network of each cell was classified as consisting of no bundles, short bundles, or long bundles. This approach gave consistent results across all three users for each dataset, so we believe it is an accurate and reliable method for assessing the F-actin organization.

4) Does anillin contribute differentially to tension/contractility and stiffness in the medial-apical network? Here it might be informative to manipulate the medial-apical region directly, by making laser cuts in the medial apical cortex itself (e.g. Stephan Grill and the 1-cell C. elegans embryo).

We thank the reviewers for suggesting that medial-apical laser cuts would allow us to further investigate the contributions of tension/contractility across the apical surface of cells. After pilot studies to determine appropriate conditions for performing medial-apical laser ablation in *Xenopus* embryos, we found that cells overexpressing Anillin exhibited increased apical expansion, while Anillin knockdown cells showed decreased apical expansion after medial-apical laser cuts. This supports the hypothesis that Anillin is promoting storage of tensile/contractile energy across the apical surface. This data is now presented in the new Figure 3C-E (see above). In addition to quantifying the perpendicular or parallel expansion after laser ablation, we also used a strain mapping technique to analyze the medial-apical laser ablation movies. This data is now presented in the new Figure 3F-G (see above).

These changes in apical expansion after medial-apical laser cuts, combined with data showing that Vinculin strongly accumulates at junctions in Anillin overexpressing cells where F-actin bundles make perpendicular connections to the junction (new Figure 3A-B), demonstrate that force generated from the medial-apical cortex is transmitted to cell-cell junctions, and that Anillin influences the orientation of these forces. This data does not exclude potential contributions from the physical stiffness of the network, and we now mention in the Discussion that in the future it would be interesting to pursue more robust stiffness measurements at the apical surface using techniques such as atomic force microscopy or micro-aspiration.

5) The reviewers wondered: Is the medial-apical network pulsatile? Is pulsatility affected by anillin overexpression (is it dampened)? If possible, measuring any pulses in the network contraction would allow the authors to add this as a part of their model for how anillin works.

We observe waves of F-actin and Myosin II that travel across the apical cortex of gastrula-stage *Xenopus* embryos similar to those observed in the early oocyte (Bement et al., 2015). These waves of F-actin appear in control, Anillin knock down, and Anillin overexpressing embryos. Perhaps, since we aren’t directly investigating a developmental event that requires apical constriction in this study, we do not observe a pulsatile, ratchet-like mechanism occurring like in *Drosophila* ventral furrow formation. We thank the reviewers for their interest in this topic, and agree that examining Anillin’s role in controlling cortical excitability at steady state and in apically constricting cells is an interesting area to explore in future studies.

6) Analysis of actin stability in the overexpression of the delta-F-actin mutant should be performed.

We agree with the reviewers, and we performed the FRAP experiments with the Anillin Δact mutant in comparison with control and Anillin FL. We confirmed our previous results that in control cells, medial-apical actin is very dynamic, and when Anillin FL is overexpressed, actin is stabilized. We found that overexpression of Anillin Δact did not significantly change the stability (mobile fraction) of medial-apical F-actin compared to controls. These data are presented in Figure 7A-C, and demonstrate that Anillin is likely stabilizing F-actin through binding to F-actin.

B) Textual changes:1) Introduction. The last two paragraphs are a little long and discursive. In particular, the final paragraph rather anticipates the story in too much detail.

We agree and have revised and reduced the amount of detail at the end of the Introduction.

2) Figure 7: morphological defects seen in presomitic mesoderm, notochord and ectoderm in KD – please describe these.

Descriptions were added for the observed defects.

C) If possible, the reviewers thought that a mechanical model could add to the interpretation of the data. They agreed that some of the experiments above may address these concerns as well. I think that it is very difficult to interpret these results without a mechanical model (or at least I find it very difficult). There are at least 3 levels of complexity that need to be considered. 1) Actomyosin can contribute to tension but also influence viscoelasticity. (Conceivably, a highly cross-linked actomyosin network might be very elastic but not actually generate much tension.) 2) As the authors note (and has been developed in some of e.g. Pierre-Francois Lenne's reviews) recoil is the product of tension divided by friction. It can't always be equated to tension; and this is potentially one of those cases, since the authors data indicate that tissue stiffness is increased. 3) Both the medial-apical and junctional networks may contribute and it is difficult to tease these apart. Looking at their videos, I would consider the possibility that the impact of anillin on the medial-apical network may be principally to increase stiffness in this region of the cell, rather than as a source of tensile force. This could explain the slower recoil of junctions when anillin is overexpressed, because a stiffer medial-apical network limits recoil of the junction. Of course, this implies that the junctions and medial-apical network are physically connected.Also the reviewers agreed that this would help the manuscript, the authors thought that if this was not able to be added in the two month time frame, the experiments requested may also address these same concerns.

We agree with the reviewers that a mechanical model could be useful to tease apart the contributions of differentially-oriented contractile tensions and network stiffness. Such models have been applied in studying apical mechanics in *Drosophila* dorsal closure where the bulk of energy storage changes from junctional apical bundles to the medial-apical cell cortex (Ma et al., 2009; Hutson et al., 2009); however, given the time frame and the need to initiate a new collaboration to develop a complex model, we felt it was not feasible at this time. The data that we have now presented in Figure 3 showing that the medial-apical network stores contractile energy and that forces are transmitted from the cortex to the junction, supports our working model Figures 5 and 9. Our findings do not exclude the possible contributions of network stiffness or friction on the cellular mechanics of these epithelial cells. Indeed, we agree that these factors are likely contributors, given that Anillin stabilizes F-actin and stiffens the tissue, and we have further emphasized this point in the Discussion.